# The tight junction protein TJP1 regulates the feeding-modulated hepatic circadian clock

Yi Liu[1,2], Yuanyuan Zhang[1,2], Tong Li[1], Jinbo Han[1] & Yiguo Wang [1]*

Circadian clocks in the suprachiasmatic nucleus and peripheral tissues orchestrate behavioral and physiological activities of mammals in response to environmental cues. In the liver, the circadian clock is also modulated by feeding. However, the molecular mechanisms involved are unclear. Here, we show that TJP1 (tight junction protein 1) functions as a mediator of mTOR (mechanistic target of rapamycin) to modulate the hepatic circadian clock. TJP1 interacts with PER1 (period circadian regulator 1) and prevents its nuclear translocation. During feeding, mTOR phosphorylates TJP1 and attenuates its association with PER1, thereby enhancing nuclear shuttling of PER1 to dampen circadian oscillation. Therefore, our results provide a previously uncharacterized mechanistic insight into how feeding modulates the hepatic circadian clock.

---

[1] MOE Key Laboratory of Bioinformatics, Tsinghua-Peking Center for Life Sciences, School of Life Sciences, Tsinghua University, Beijing 100084, China. [2] These authors contributed equally: Yi Liu, Yuanyuan Zhang. *email: yiguo@mail.tsinghua.edu.cn

The circadian clock is an endogenous timekeeper that coordinates behavior, physiology, and metabolism in a wide range of organisms[1,2]. In mammals, the central pacemaker in the suprachiasmatic nucleus (SCN) synchronizes oscillations in peripheral organs in response to light–dark cycles[1,2]. The molecular clock is driven by interconnected transcriptional feedback loops involving both transcriptional activators and repressors to produce self-sustained rhythmic transcription of target genes[1–3]. In this process, a heterodimer of the transcription factors CLOCK (circadian locomotor output cycles kaput) and BMAL1 (brain and muscle ARNT-like 1) acts as a transcriptional activator that binds to E-box motifs in target genes and drives their transcription[1–3]. These target genes include period (Per), cryptochrome (Cry), and Rev-Erb, which encode repressors of CLOCK and BMAL1. PER and CRY rhythmically accumulate, shuttle to the nucleus and form a repressor complex that interacts with CLOCK:BMAL1 to inhibit transcription of Per and Cry[1–3]. Another important diurnal loop involves the transcriptional repressor REV-ERB and a member of the ROR (retinoid-related orphan receptor) family of transcriptional activators, which together control the rhythmic expression of BMAL1[1–3]. Unlike the SCN clock, the circadian clock in the liver is also modulated by feeding, such as refeeding after fasting, restricted feeding, and high fat diet feeding, to coordinate energy metabolism[2,4–14]. However, the molecular mechanisms by which feeding affects the circadian clock remain unclear.

The liver, which is composed of polarized hepatocytes and other cells[15,16], is one of the most critical metabolic organs in mammals. The polarization of hepatocytes involves formation of distinct sinusoidal and bile canalicular plasma membrane domains that are separated by tight junctions[15,16]. These tight junctions form the blood–biliary-barrier, which keeps bile in bile canaliculi away from the blood circulation and also acts as a signaling platform for communicating between the inside and the outside of the cells[15,16]. Although tight junction proteins are critical to maintain liver function, it is unknown whether they modulate the hepatic circadian clock.

In this study, we demonstrated that tight junction protein 1 (TJP1) functions as a mediator of mTOR to modulate the hepatic circadian clock. Mechanistically, feeding activates mTOR, which phosphorylates TJP1, thereby disrupting the TJP1:PER1 association and promoting PER1 nuclear translocation to inhibit the expression of CLOCK:BMAL1 target genes. This discovery provides a mechanistic insight into how feeding modulates the circadian clock in the liver and links the junction complex to circadian rhythm.

## Results

**Tjp1 knockout suppresses hepatic circadian amplitude**. To maintain the characteristics of hepatocytes in vitro, we cultured mouse primary hepatocytes in a collagen sandwich configuration[17,18] and evaluated the bile canaliculus-like structures by immunostaining of CLDN1 and CGN, which are tight junction markers (Supplementary Fig. 1a). These results showed that hepatocytes cultured in a collagen sandwich (SD+) have a clear bile canaliculus-like structure, similar to the previous reports[18]. Interestingly, SD+ culture dramatically enhanced the expression of Rev-Erbα and circadian amplitude (Fig. 1a). Strikingly, knockout of Tjp1 (encoding tight junction protein 1) strongly repressed Rev-Erbα expression and circadian amplitude (Fig. 1a and Supplementary Fig. 1b–d), while having no effect on tight junction formation evaluated by CLDN1 and CGN staining (Supplementary Fig. 1a) and measurement of transepithelial electrical resistance (TER, Supplementary Fig. 1e), which reflects paracellular permeability regulated by the tight junction[19].

Previous reports also showed that the knockout of Tjp1 in epithelial cells did not affect tight junction formation[20,21].

We further tested the effect of TJP1 on the circadian clock in Tjp1 liver-specific knockout (LKO) mice. Tjp1 LKO mice have similar tight junction structures and bile acid levels to wildtype ones (Supplementary Fig. 1f–h). However, Tjp1 deficiency notably repressed the expression of E-box-containing genes (Rev-Erbα, Dbp, and Per1) in the liver, but not in the SCN, and increased the nuclear translocation of PER1 and CRY1, but not PER2, in the liver (Fig. 1b–g and Supplementary Fig. 1i, j). Since liver-specific Tjp1 knockout has no effect on the circadian gene expression of SCN, these results show that TJP1 autonomously modulates circadian amplitude in the liver.

**mTOR attenuates the association of TJP1 and PER1**. Next, we tested how TJP1 affects PER1/CRY1 nuclear translocation. We identified TJP1-interacting proteins by immunoprecipitation (IP) of endogenous TJP1 in SD+ cultured primary hepatocytes and mass spectrometry (MS) analysis. We found that both PER1 and CRY1 were present in the TJP1 immunoprecipitates (Supplementary Fig. 2a). Co-IP assay showed that TJP1 is associated with both wildtype PER1 and the binding-defective mutant of PER1 (1–1147 aa) on CRY1 (Supplementary Fig. 2b), indicating that PER1 but not CRY1 interacts with TJP1. In addition, TJP1 is not associated with other circadian core components, such as BMAL1 and CLOCK (Supplementary Fig. 2c). Further analysis by in vitro and in vivo co-IP assays showed that the C-terminus (511–1748 aa) of TJP1 is responsible for the TJP1:PER1 interaction (Supplementary Fig. 2d, e).

Recent evidence has shown that mTOR affects the circadian clock in cultured cells, SCN, and peripheral organs[22–28]. However, the mechanisms by which mTOR modulates the circadian clock are largely unclear. We tested whether mTOR regulates the TJP1:PER1 association to modulate the circadian clock. Strikingly, addition of amino acids (AA) to activate mTOR, decreased the TJP1:PER1 association at the lysosome and promoted PER1 nuclear translocation, while addition of torin1, an mTOR inhibitor, abolished the effect of AAs (Fig. 2a–c and Supplementary Fig. 3a–c). mTOR does not affect the localization of TJP1 in lysosomal fractions or at junctions (Supplementary Fig. 3c, d). This indicates that mTOR promotes the dissociation of TJP1 and PER1 at lysosomal fractions. In addition, the TJP1:PER1 association was dynamically regulated in a manner related to mTOR activity in mice. There was lower mTOR activity and stronger binding of TJP1 to PER1 at ZT6, and vice versa at ZT18 (Fig. 2d and Supplementary Fig. 3e).

Although mTOR promotes PER1 nuclear translocation, neither PER1 nor CRY1 was phosphorylated by mTOR (Supplementary Fig. 3f–h). Activated mTOR had a stronger binding to TJP1 (Fig. 2e and Supplementary Fig. 3i, j). We therefore identified the preferred mTOR phosphorylation motifs in TJP1[29] and checked them by amino acid scanning. Alanine or aspartic acid mutations of six conserved AA affected the binding of TJP1 to PER1 and thus were judged to be critical for the TJP1:PER1 association (Supplementary Fig. 3k, l). We further confirmed that mTOR phosphorylates TJP1 by an in vitro kinase assay (Fig. 2f). Phospho-specific antibodies indicated that phosphorylation of TJP1 is tightly modulated by mTOR both in cultured hepatocytes and in mice (Fig. 2a, d and Supplementary Fig. 3m). The phosphorylation-defective mutant of TJP1 (6A) constitutively interacted with PER1 and almost lost its response to AA stimulation, while the phosphorylation-mimic mutant of TJP1 (6D) did not bind to PER1 (Fig. 2g). Taken together, these results indicate that mTOR activation stimulated by AA attenuates the TJP1:PER1 association by phosphorylating TJP1.

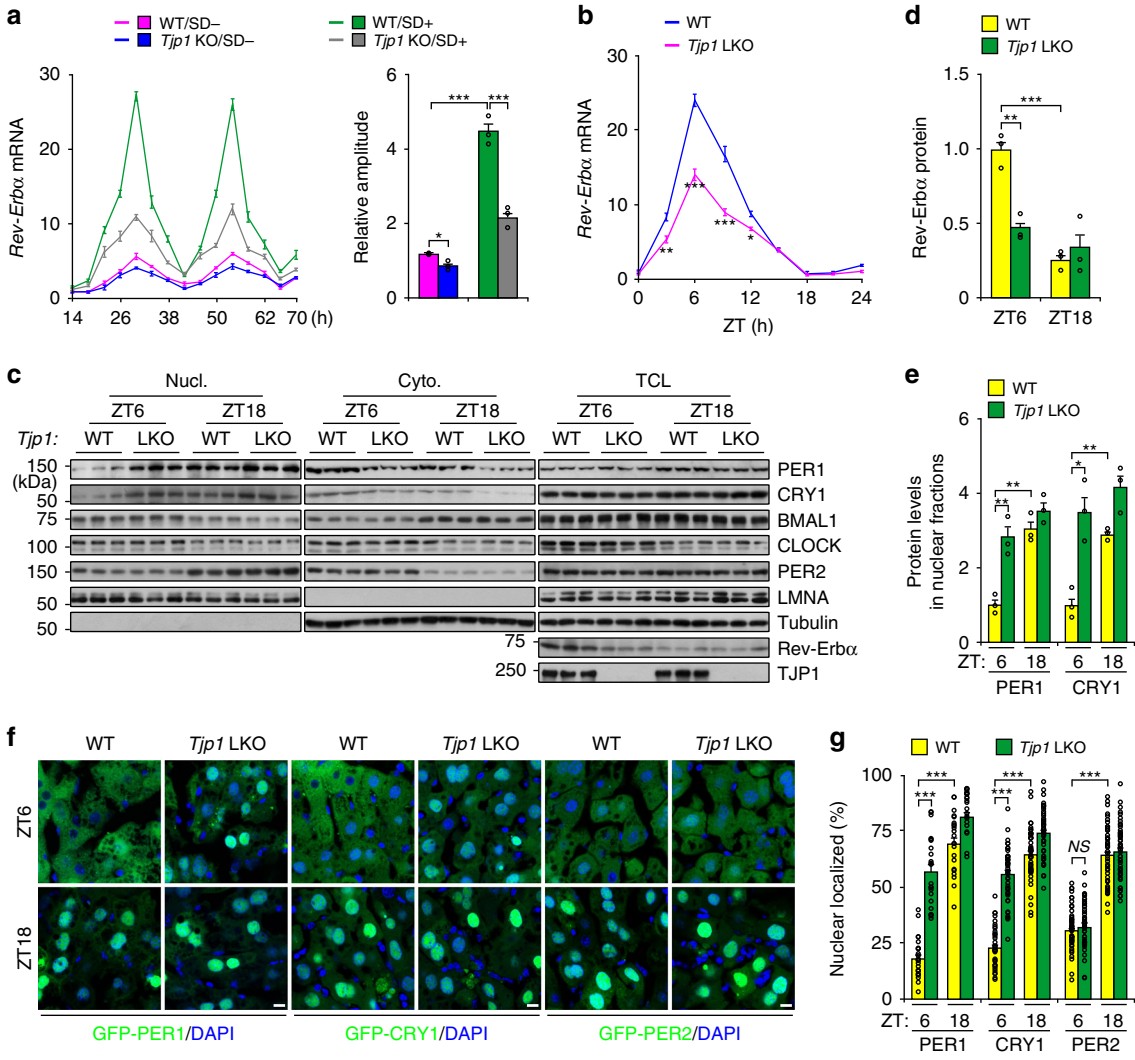

**Fig. 1 *Tjp1* deficiency suppresses circadian amplitude in the liver. a** Relative mRNA level of *Rev-Erbα* (left panel) and rhythmic amplitude (right panel) in wildtype (WT) or *Tjp1* liver-specific knockout (LKO) mouse primary hepatocytes cultured in the presence (SD+) or absence (SD−) of a collagen sandwich configuration. Hepatocytes were exposed to dexamethasone (0.1 μM) and then harvested at different time points. $n = 6$. **b** qPCR results showing the expression of *Rev-Erbα* in liver extracts from WT and *Tjp1* LKO mice. $n = 3-7$ mice. **c** Immunoblots showing the levels of core circadian components in different fractions (Nucl. for nuclear fraction, Cyto. for cytosolic fraction, and TCL for total cell lysate) of liver extracts from WT and *Tjp1* LKO mice. **d** and **e** Statistical analysis of Rev-Erbα in TCL **d**, and PER1 and CRY1 in Nucl. **e** From immunoblots as shown in **c**. $n = 6$ mice. **f** and **g** Cellular localization of GFP-PER1, GFP-CRY1, or GFP-PER2 **f** and statistical analysis of the results **g** showing the percentage of nuclear-localized GFP-PER1, GFP-CRY1, or GFP-PER2 in the liver from WT and *Tjp1* LKO mice at ZT6 and ZT18. Nuclei are stained by 4,6-diamidino-2-phenylindole (DAPI). $n = 6$ mice. Scale bars, 10 μm. ZT is the *Zeitgeber* time. Data are shown as mean ± s.e.m. Comparison of different groups was carried out using two-way ANOVA. *$P < 0.05$, **$P < 0.01$, ***$P < 0.001$. NS, no statistical significance. Source data are provided as a Source Data file.

**Phosphorylation of TJP1 promotes nuclear shuttling of PER1.** Since the phosphorylation of TJP1 attenuates the TJP1:PER1 association, we checked whether the phosphorylation of TJP1 affects PER1 nuclear translocation. The nuclear translocation of PER1 stimulated by AA in primary hepatocytes was blocked by the phosphorylation-defective mutant of TJP1 (6A), while TJP1 6D constitutively promotes nuclear translocation of PER1 (Supplementary Fig. 4). We further confirmed these effects by adenoviral expression of mCherry-tagged TJP1 in *Tjp1* LKO mice. Nuclear translocation of PER1 was blocked and expression of *Rev-Erbα* was enhanced in these animals (Fig. 3a–f). In addition, PER1 nuclear translocation in *Tjp1* LKO mice was blocked by the mTOR-defective mutant of TJP1 (6A) but not the mTOR-mimic mutant (6D), which further supports the notion that phosphorylation of TJP1 by mTOR is critical for PER1 nuclear shuttling (Fig. 3a–f). Together, these results indicate that mTOR promotes

nuclear translocation of PER1 by phosphorylating TJP1 and disrupting its association with PER1.

**mTOR dampens hepatic circadian amplitude.** Considering the effect of phosphorylation of TJP1 by mTOR on PER1 nuclear shuttling, we checked whether mTOR itself modulates the hepatic circadian clock. To address this, we interrupted mTOR activity by Torin1 treatment or by deleting the gene encoding RAPTOR, a subunit of mTOR complex 1 (mTORC1), and we enhanced mTOR activity by deleting the gene encoding TSC1, a GTPase-activating protein that inhibits mTORC1 activity[30]. In the absence of mTOR manipulations, *Rev-Erbα* expression, mTOR activity, and pTJP1 levels are dramatically enhanced in SD+ cultured hepatocytes compared to SD− cultured hepatocytes (Supplementary Fig. 5a–c). However, Torin1 treatment further

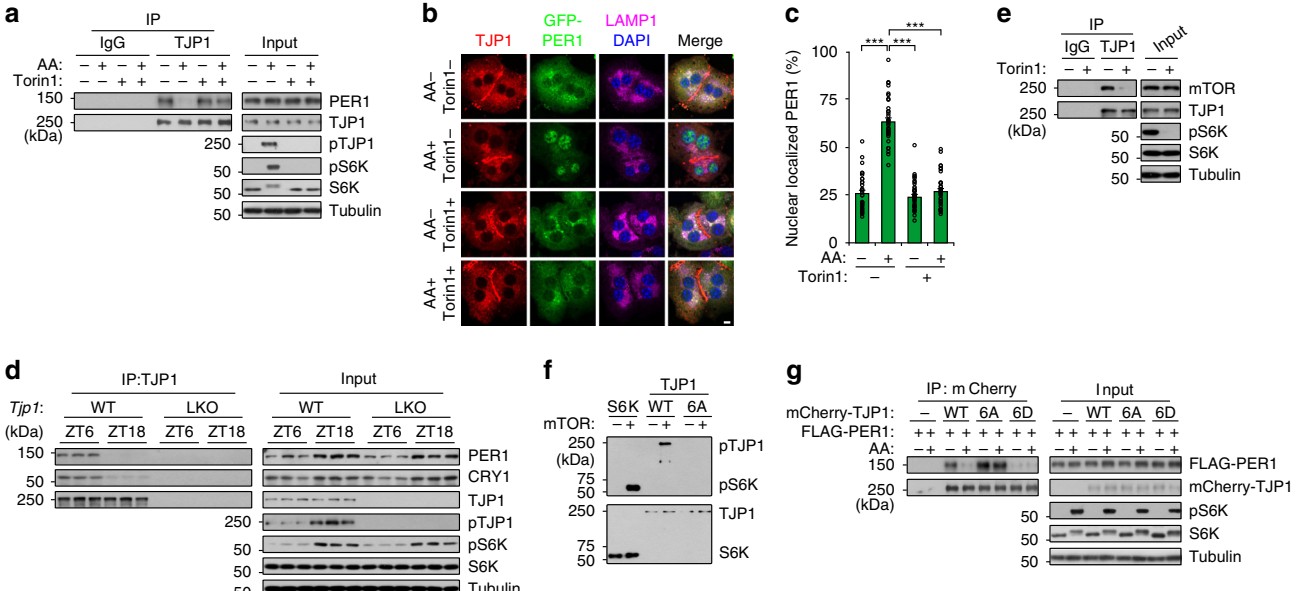

**Fig. 2 mTOR attenuates the association of TJP1 and PER1. a** Co-immunoprecipitation (co-IP) showing the interaction of endogenous TJP1 and PER1 in primary hepatocytes cultured in a collagen sandwich configuration in response to treatment with amino acids and/or Torin1. Mouse primary hepatocytes incubated with amino acid-free RPMI1640 for 6 h were exposed to 250 nM Torin1 or control vehicle for another 30 min, then treated with amino acids for 30 min. **b** and **c** Cellular localization of GFP-PER1 and endogenous TJP1 **b** and statistical analysis of the results **c** in primary hepatocytes cultured in a collagen sandwich configuration in response to treatment with amino acids and/or Torin1. Mouse primary hepatocytes incubated with amino acid-free RPMI1640 for 6 h were exposed to 250 nM Torin1 or control vehicle for another 30 min, then treated with amino acids for 30 min. Scale bar, 10 μm. Data are shown as mean ± s.e.m. Comparison of different groups was carried out using two-way ANOVA. ***$P < 0.001$, $n = 6$. **d** Immunoblots of co-IP assays showing the interaction of endogenous TJP1 and PER1 in liver extracts harvested from mice at ZT6 and ZT18. **e** Co-IP assay showing the interaction of endogenous mTOR and TJP1 in primary hepatocytes cultured in a collagen sandwich configuration. Mouse primary hepatocytes were exposed to 250 nM Torin1 or control vehicle for 1 h. **f** In vitro kinase assay showing the phosphorylation of HA-tagged TJP1 and S6K by truncated mTOR. HA-tagged TJP1 and S6K were purified from mouse *Mtor* LKO primary hepatocytes infected with adenovirus expressing HA-tagged S6K or wildtype TJP1 or mutant TJP1 (6A). **g** Immunoblots of co-IP assays showing the interaction of FLAG-PER1 and mCherry-tagged TJP1 or its mutants in mouse primary hepatocytes in the presence or absence of amino acids. 6A is the T589A/T709A/S927A/S1360A/S1460A/S1617A mutant of human TJP1, while 6D is the T589D/T709D/S927D/S1360D/S1460D/S1617D mutant of human TJP1. pTJP1 indicates human TJP1 phosphorylated at S1617 or mouse TJP1 phosphorylated at S1614. Source data are provided as a Source Data file.

enhanced both *Rev-Erbα* expression and circadian amplitude (Supplementary Fig. 5a, b). In addition, *Raptor* deficiency abolished mTORC1 activity and increased both *Rev-Erbα* expression and circadian amplitude, while *Tsc1* deficiency enhanced mTORC1 activity and decreased *Rev-Erbα* expression and circadian amplitude (Fig. 4a and Supplementary Fig. 5d–f). These results were further confirmed by data showing that PER1 nuclear translocation was repressed and *Rev-Erbα* expression was enhanced in *Raptor* LKO mice (Fig. 4b–g and Supplementary Fig. 5g). Together, these results showed that mTOR modulates the circadian clock in the liver.

**Feeding enhances PER1 nuclear shuttling via mTOR and TJP1.** Feeding in mice mostly occurs at night and constitutes around 80% of daily food intake[14]. In addition, feeding is a critical factor for activating mTOR and modulating the hepatic circadian clock[5–12,31,32]. Considering these facts and our results, we hypothesized that feeding activates mTOR and thereby modulates the circadian clock by phosphorylating TJP1 to affect PER1 nuclear translocation. We fasted mice from ZT12 to ZT4 and then refed them for another 2 h to test this hypothesis. As shown in Fig. 5, feeding promoted PER1 nuclear translocation in WT mice, but not in *Raptor* LKO mice. In contrast to *Raptor* LKO mice, PER1 showed obvious nuclear translocation even under fasting in both *Tsc1* LKO and *Tjp1* LKO mice (Fig. 5a–e). Correspondingly, *Rev-Erbα* expression was suppressed by PER1 nuclear translocation (Fig. 5f, g). In summary, these results

showed that both TJP1 and mTOR are critical for the feeding-modulated nuclear translocation of PER1 in the liver.

**An mTOR–TJP1 axis regulates circadian clock during feeding.** We next investigated whether the specific function of TJP1 as a downstream mediator of mTOR is important for the feeding-modulated hepatic circadian clock. Feeding-promoted nuclear translocation of PER1 was blocked by *Raptor* deficiency, while *Raptor* and *Tjp1* double knockout (DKO) in the liver restored nuclear translocation of PER1 and inhibited *Rev-Erbα* expression (Fig. 6a–f). Although mTOR is less active during fasting, *Tsc1* deficiency constitutively activated mTOR, promoted PER1 nuclear translocation, and reduced *Rev-Erbα* expression (Fig. 6g–l). These effects were blocked by the mTOR-defective mutant of TJP1 (6A) (Fig. 6g–l). Together, these results indicate that mTOR modulates the hepatic circadian clock by phosphorylating TJP1 during feeding.

**TJP1 deficiency improves insulin sensitivity.** Altered circadian clock is tightly associated with insulin sensitivity, a hallmark of diabetes[1,2,4,33]. Considering the effect of TJP1 on the hepatic circadian clock, we tested whether TJP1 affects mouse insulin sensitivity. *Tjp1* knockout mice fed with a regular diet (RD) showed enhanced nuclear translocation of PER1, and reduced *Rev-Erbα* expression, blood glucose level, plasma insulin level, plasma triglyceride (TG) level, and hepatic TG accumulation (Fig. 7a–j). However, *Tjp1* knockout mice fed a RD had a similar body weight,

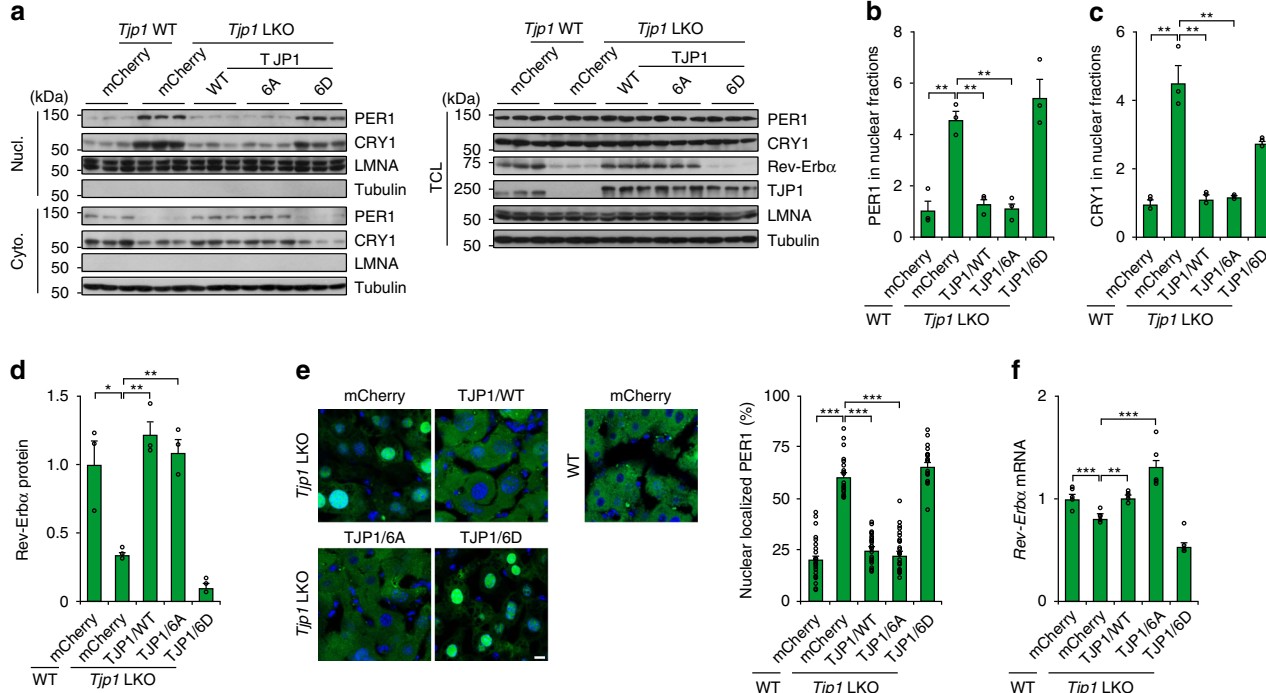

**Fig. 3 Phosphorylation of TJP1 promotes PER1 nuclear translocation. a–d** Effect of TJP1 or its mutants on cellular localization of PER1 in the liver evaluated by immunoblots **a**, and statistical analysis of PER1 **b**, CRY1 **c** in Nucl. and Rev-Erbα in TCL **d** of liver extracts harvested from mice at ZT6. **e** and **f** Effect of TJP1 or its mutants on cellular localization of GFP-PER1 **e**, and *Rev-Erbα* expression **f** in liver extracts harvested from mice at ZT6. 6A is the T589A/T709A/S927A/S1360A/S1460A/S1617A mutant of human TJP1, while 6D is the T589D/T709D/S927D/S1360D/S1460D/S1617D mutant of human TJP1. Data are shown as mean ± s.e.m. Comparison of different groups was carried out using one-way ANOVA. *$P < 0.05$, **$P < 0.01$, ***$P < 0.001$, $n = 5$ mice. Source data are provided as a Source Data file.

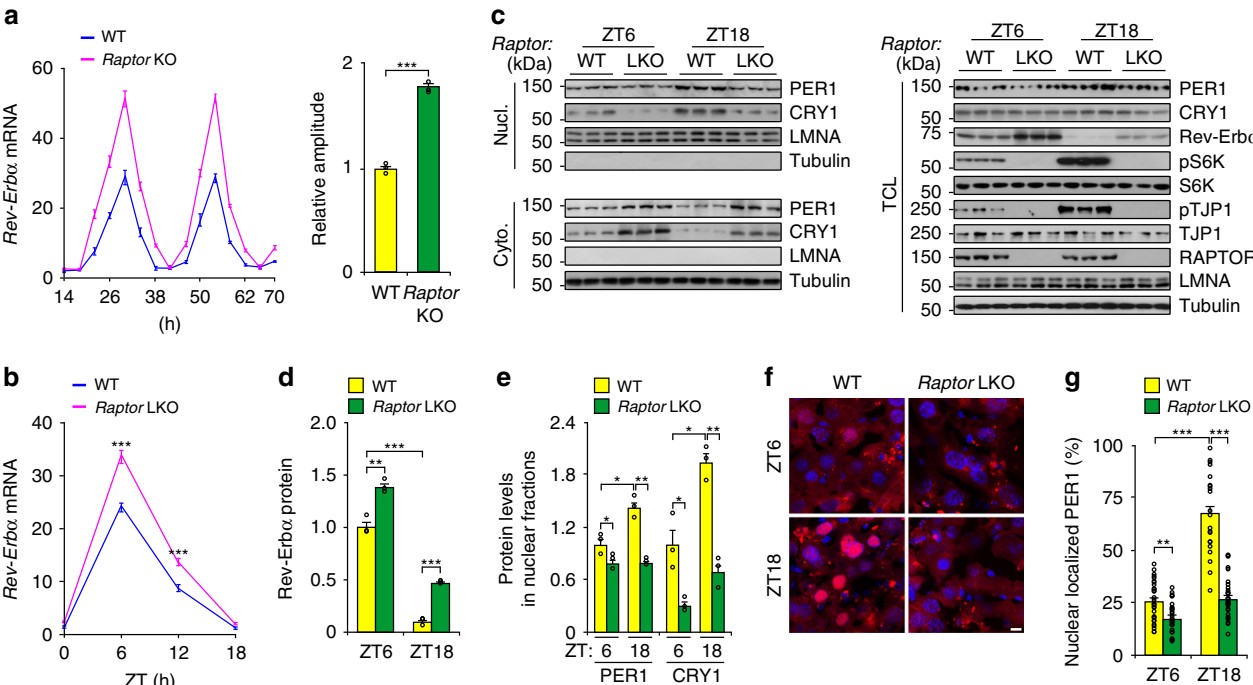

**Fig. 4 mTOR dampens hepatic circadian amplitude. a** Relative mRNA level of *Rev-Erbα* (left panel) and rhythmic amplitude (right panel) in wildtype (WT) or *Raptor* LKO mouse primary hepatocytes cultured in a collagen sandwich configuration. Hepatocytes were exposed to dexamethasone (0.1 μM) and then harvested at different time points. $n = 6$. **b** Effect of *Raptor* LKO on *Rev-Erbα* expression in the liver from mice at ZT6 and ZT18. $n = 4$–7 mice. **c–e** Effect of *Raptor* LKO on cellular localization of PER1 in the liver evaluated by immunoblots **c**, and statistical analysis of Rev-Erbα in TCL **d**, and PER1 and CRY1 in Nucl. **e** from immunoblots as shown in **c**. $n = 4$–7 mice. **f** and **g** Effect of *Raptor* LKO on cellular localization of PER1 in the liver visualized by mCherry-PER1 **f** and statistical analysis of the results **g**. $n = 4$–7 mice. Scale bar, 10 μm. Data are shown as mean ± s.e.m. Comparison of different groups was carried out using two-tailed unpaired Student's $t$-test **a** or two-way ANOVA **b**, **d**, **e** and **g**. *$P < 0.05$, **$P < 0.01$, ***$P < 0.001$. Source data are provided as a Source Data file.

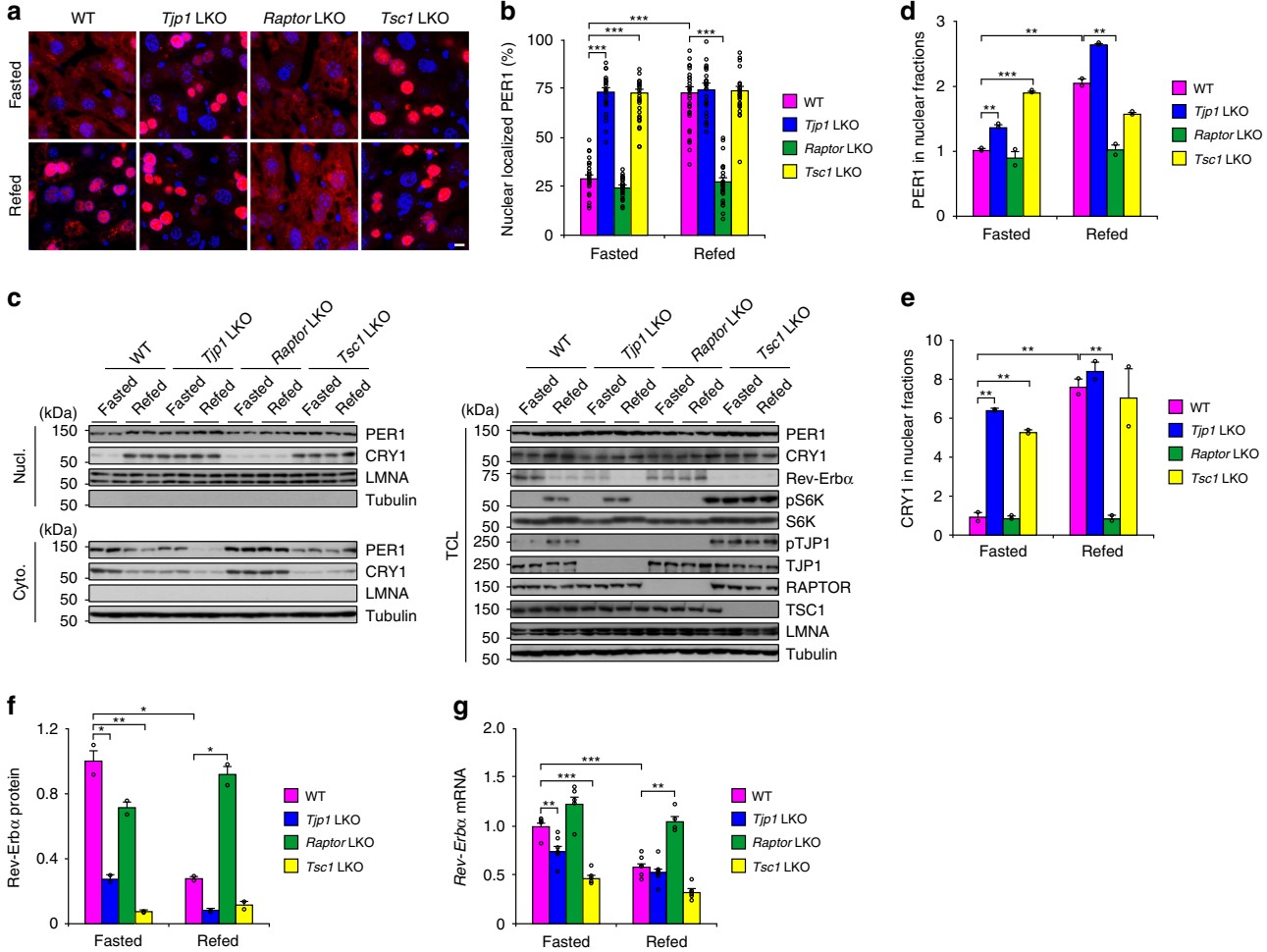

**Fig. 5 TJP1 and mTOR mediate feeding-induced nuclear translocation of PER1. a and b** Cellular localization of mCherry-PER1 **a** and the statistical analysis of nuclear-localized mCherry-PER1 **b** in the liver from mice fasted from ZT12 to ZT4 or refed for another 2 h from ZT4 to ZT6. **c** Immunoblots showing amounts of PER1 in nuclear fractions of liver extracts from mice fasted from ZT12 to ZT4 or refed for another 2 h from ZT4 to ZT6. **d–f** Statistical analysis of PER1 **d** and CRY1 **e** in Nucl. and Rev-Erbα in TCL **f** from immunoblots as shown in **c**. **g** qPCR results showing the relative mRNA level of *Rev-Erbα* in liver extracts from mice fasted from ZT12 to ZT4 or refed for another 2 h from ZT4 to ZT6. Data are shown as mean ± s.e.m. Comparison of different groups was carried out using two-way ANOVA. *$P < 0.05$, **$P < 0.01$, ***$P < 0.001$, $n = 5$–7 mice. Source data are provided as a Source Data file.

fat mass, food intake, energy expenditure, circadian gene expression, and cholesterol levels to wildtype ones (Supplementary Fig. 6a–j). To evaluate the effect of TJP1 on insulin sensitivity, we performed hyperinsulinemic-euglycemic clamp studies. Compared to WT mice, the steady-state glucose infusion rate (GIR) in *Tjp1* LKO mice was dramatically increased, reflecting enhanced whole-body insulin responsiveness, and was accompanied by an increase in the glucose disposal rate (GDR, Fig. 7k, l). *Tjp1* LKO mice showed marked increases in the insulin-induced hepatic glucose production (HGP) suppression, which reflects liver insulin sensitivity; the insulin-stimulated GDR (IS-GDR), which primarily reflects skeletal muscle insulin sensitivity; and the insulin-induced suppression of plasma FFA levels, which indicates white adipose tissue insulin sensitivity (Fig. 7m–o).

Since mice fed with a high-fat diet (HFD) showed a dampened hepatic circadian amplitude, enhanced mTOR activity, and attenuated insulin sensitivity[14,31], we tested whether *Tjp1* deficiency can improve insulin sensitivity in HFD-fed mice. Compared to WT mice fed with a HFD for 16 weeks, *Tjp1* LKO mice showed enhanced PER1/CRY1 nuclear shuttling, an improved metabolic profile and restored insulin sensitivity (Fig. 7 and Supplementary Fig. 6). All these results demonstrate that TJP1 links hepatic circadian clock to insulin sensitivity.

## Discussion

Although it is well known that the circadian clock in the liver is modulated by feeding to coordinate energy metabolism[2,4–14], the molecular mechanisms by which feeding modulates circadian clock remain unclear. Our results demonstrate that feeding activates mTOR and thereby phosphorylates TJP1, which disrupts the TJP1:PER1 association and thus promotes PER1/CRY1 nuclear translocation to inhibit the expression of CLOCK:BMAL1 target genes (Fig. 8). This discovery provides mechanistic insight into how feeding modulates the circadian clock in the liver and links the junction complex to circadian rhythm.

Since the circadian amplitude is affected in obese or diabetic models, modulating circadian amplitude can combat the negative effects of obesity[14,34–37]. In support of this notion, *Tjp1* deficiency restores the HFD-induced effect on circadian gene expression, metabolic profile, and insulin sensitivity (Fig. 7), although it is possible that TJP1 can also improve insulin sensitivity in a clock-independent manner. Therefore, our results demonstrate a potential molecular link between circadian amplitude and insulin resistance.

Activated mTOR phosphorylates TJP1 and disrupts the colocalization of TJP1 and PER1/CRY1 at lysosomes, thereby promoting the nuclear translocation of PER1 (Fig. 2b and

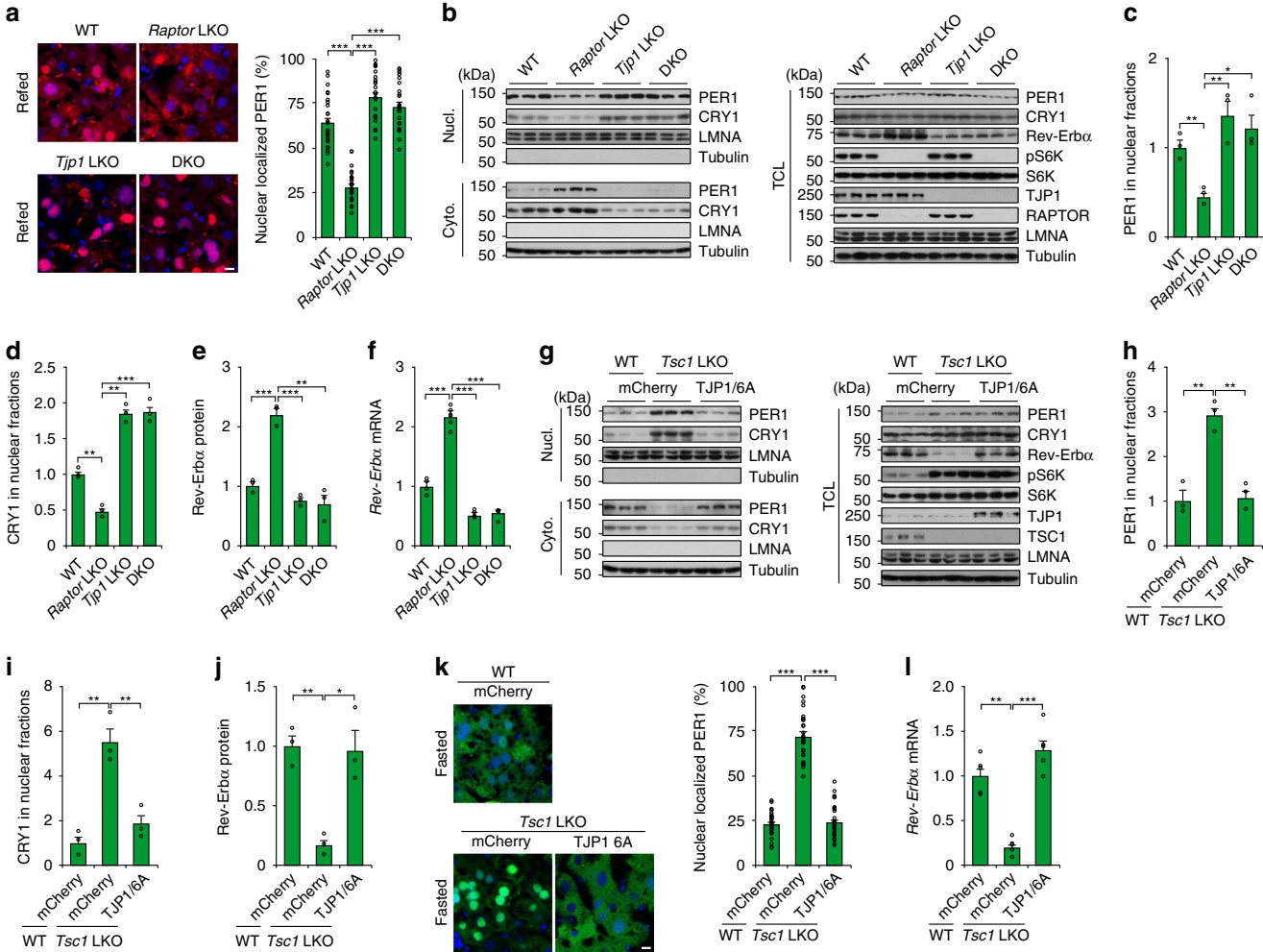

**Fig. 6 TJP1 regulates the feeding-induced effect of mTOR on the circadian clock. a and b** Cellular localization of mCherry-PER1 in the liver **a** and immunoblots showing the amount of PER1 **b** in liver extracts from WT, *Raptor* LKO, *Tjp1* LKO, and *Raptor/Tjp1* double knockout (DKO) fed mice. $n = 4$–6 mice. **c–e** Statistical analysis of PER1 **c** and CRY1 **d** in Nucl. and Rev-Erbα in TCL **e** from immunoblots as shown in **b**. $n = 4$–6 mice. **f** qPCR results showing *Rev-Erbα* expression in liver extracts from WT, *Raptor* LKO, *Tjp1* LKO, and *Raptor/Tjp1* double knockout (DKO) fed mice. $n = 4$–6 mice. **g–j** Effect of *Tsc1* LKO and addition of TJP1/6A on PER1 level in the nuclear fraction **g**, and statistical analysis of PER1 **h** and CRY1 **i** in Nucl. and Rev-Erbα in TCL **j** from immunoblots as shown in **g** in liver extracts from WT, *Tsc1* LKO, and mCherry-TJP1/6A-overexpressing mice after overnight fasting. $n = 5$ mice. **k and l** Effect of *Tsc1* LKO and addition of TJP1/6A on cellular localization of GFP-PER1 in the liver **k** and *Rev-Erbα* expression **l** in liver extracts from WT, *Tsc1* LKO, and mCherry-TJP1/6A-overexpressing mice after overnight fasting. $n = 5$ mice. Scale bars, 10 μm. Data are shown as mean ± s.e.m. Comparison of different groups was carried out using one-way ANOVA. *$P < 0.05$, **$P < 0.01$, ***$P < 0.001$. Source data are provided as a Source Data file.

Supplementary Fig. 3a). It is unclear how phosphorylation of TJP1 affects the TJP1:PER1 interaction. Our results showed that the mTOR phosphorylation sites in TJP1 localize to the region of TJP1 that interacts with PER1, and some of these phosphorylation sites are also located in the protein domains of TJP1. Based on previous reports[38,39], it is possible that the phosphorylation of TJP1 modulates TJP1:PER1 binding by directly disrupting the interaction or by changing the conformation of TJP1. More detailed structural information is needed to address this issue.

The lysosome is a central organelle in energy metabolism and nutrient sensing and recycling[40–42]. Recruitment of mTOR to lysosomes is required for mTOR activation, while amino acid stimulation enhances mTOR accumulation and activation at lysosomes[30] (Supplementary Fig. 3c, i and 5c). It is reported that lysosome-dependent redistribution and degradation of TJP1 contributes to the hyperglycemia-induced increase in the permeability of the blood–brain barrier[43]. Although the function of lysosome-localized TJP1 and PER1/CRY1 is unclear, it is possible that lysosomes provide an interface to link nutrient status to the circadian clock. Consistent with this notion, autophagy promotes lysosome-dependent degradation of CRY1 and glucose metabolism during fasting[44].

Previous studies showed that casein kinase phosphorylates PER1 and CRY1 to promote their nuclear shuttling, while AMPK, an energy stress sensor, phosphorylates CRY1 and enhances its ubiquitin-dependent degradation[45–48]. Our results showed that mTOR promotes the nuclear translocation of PER1/CRY1. All the results reflect the fact that different nutrient and energy statuses or metabolic conditions are tightly coupled to the circadian clock. Considering the ubiquitous expression of mTOR and TJP1, it will be interesting in the future to test whether the mTOR–TJP1 signaling axis also affects the circadian clock in other organs or non-polarized cells.

## Methods
**Mouse strains.** Mice carrying FRT and a floxed allele of *Tjp1* (*Tjp1*fl/+) were obtained from International Mouse Phenotyping Consortium (IMPC), and then crossed with mice expressing the FLP recombinase to obtain *Tjp1*fl/+ mice. *Tjp1*fl/+

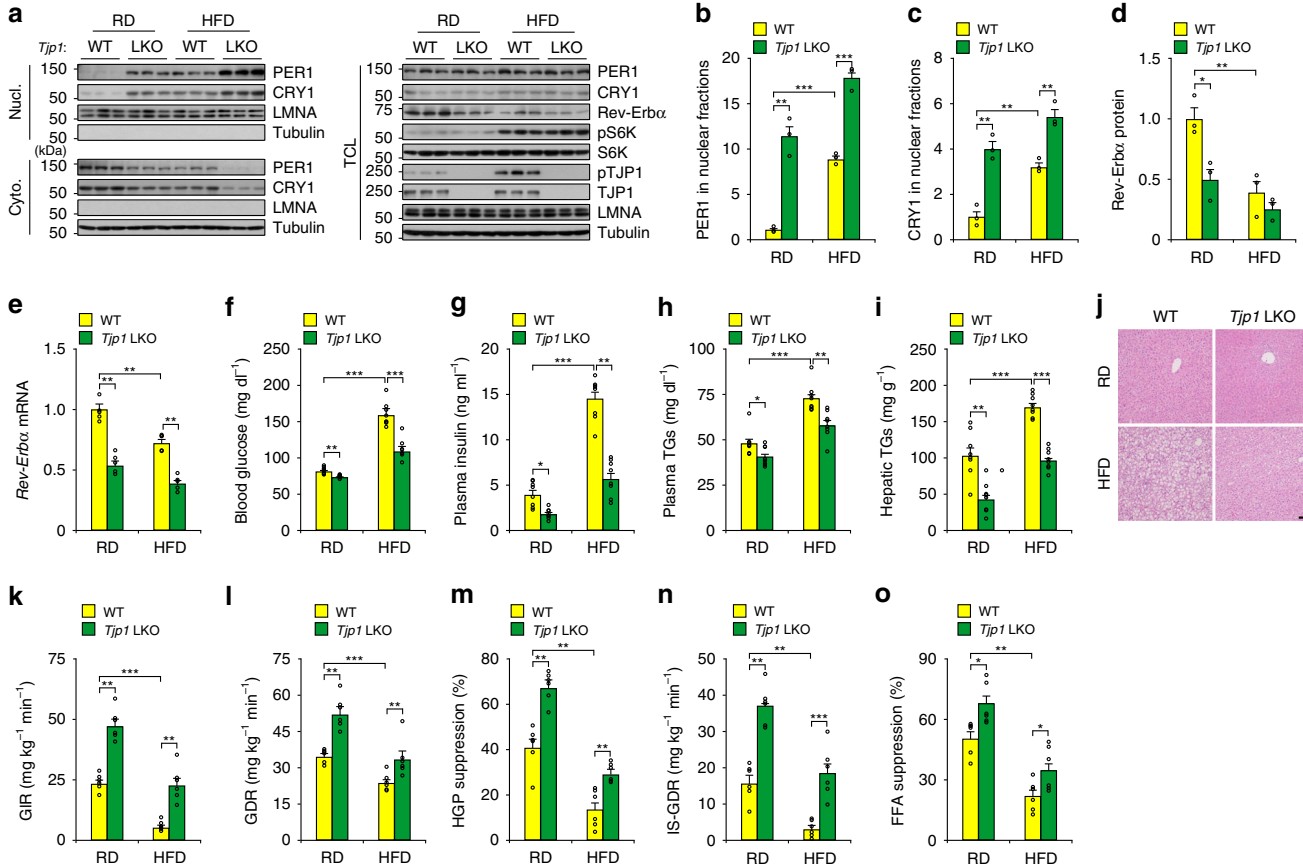

**Fig. 7 TJP1 deficiency enhances insulin sensitivity. a–e** Effect of *Tjp1* LKO on nuclear PER1 levels evaluated by immunoblots **a**, statistical analysis of PER1 **b**, and CRY1 **c** in Nucl., statistical analysis of Rev-Erbα in TCL **d**, and *Rev-Erbα* expression **e** in liver extracts from mice fed on a RD or HFD for 16 weeks. *n* = 5 mice. **f–i** Effect of *Tjp1* LKO on blood glucose **f**, plasma insulin levels **g**, plasma triglyceride levels **h** and hepatic triglyceride contents **i** from mice fed on a RD or HFD for 16 weeks. TGs triglycerides. *n* = 8 mice. **j** Hematoxylin and eosin staining of liver from WT and *Tjp1* LKO mice fed on a RD or HFD for 16 weeks. Scale bar, 50 μm. **k–o** Glucose infusion rate (GIR, **k**), glucose disposal rate (GDR, **l**), percentage suppression of HGP **m**, insulin-stimulated GDR (IS-GDR, **n**) and percentage of free fatty acid (FFA) suppression **o** from WT and *Tjp1* LKO mice fed on a RD or HFD for 16 weeks. *n* = 6 mice. Data are shown as mean ± s.e.m. Comparison of different groups was carried out using two-way ANOVA. *$P < 0.05$, **$P < 0.01$, ***$P < 0.001$. Source data are provided as a Source Data file.

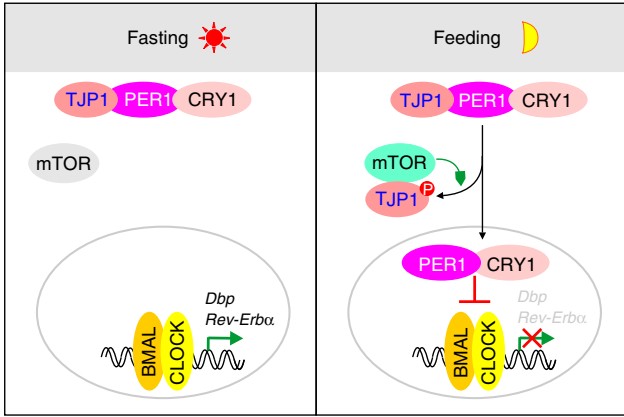

**Fig. 8 TJP1 couples mTOR signaling to the hepatic circadian clock.** During fasting or in daytime, mTOR is less active and cannot disrupt the association of TJP1 and PER1/CRY1, which sequesters PER1/CRY1 in the cytoplasm. During feeding or at nighttime, active mTOR phosphorylates and interacts with TJP1, disrupts its interaction with PER1/CRY1 and promotes nuclear translocation of PER1/CRY1, which inhibits CLOCK: BMAL-dependent transcription.

mice were backcrossed to C57BL/6 mice for six generations, and then bred to homozygosity. For generation of liver-specific *Tjp1* knockout (*Tjp1* LKO) mice, *Tjp1*^fl/fl mice were crossed with mice expressing the Cre-recombinase transgene from the liver-specific albumin promoter (*Alb*-Cre mice) to delete exon 3. Study cohorts were generated by crossing *Tjp1*^fl/fl (WT) mice with *Alb*-Cre *Tjp1*^fl/fl (*Tjp1* LKO) mice. PCR genotyping of *Tjp1* LKO mice was performed with primers that detect the following: 3′ LoxP site of the targeted *Tjp1* allele: forward, ACCCTGTACTGTACTTAGGTAAT; reverse, ACTCTCCTGTGAAAGGCCCT CAA. *Alb*-Cre: forward, GCCTGCATTACCGGTCGATGC; reverse, CAGGG TGTTATAAGCAATCCC. *Tsc1*^fl/fl (005680), *Raptor*^fl/fl (013188), and *Mtor*^fl/fl (011009) mice were from Jackson Laboratory (kindly provided by Dr. Suneng Fu, Tsinghua University) and bred to homozygosity. For generation of liver-specific *Tsc1* knockout (*Tsc1* LKO) mice, *Tsc1*^fl/fl mice were crossed with *Alb*-Cre mice. PCR genotyping of *Tsc1* LKO mice was performed with the following primers to detect the 3′ LoxP site of the targeted *Tsc1* allele: forward, GTCACGACCGTA GGAGAAGC; reverse, GAATCAACCCCACAGAGCAT. To generate the liver-specific *Raptor* knockout (*Raptor* LKO) mice, high-titer adenoviral-Cre (1 × 10^9 pfu) was delivered to *Raptor*^fl/fl mice by tail vein injection, and subsequent experiments were performed within 2 weeks. PCR genotyping for the floxed allele was performed with the following primers to detect the 3′ LoxP site of the targeted *Raptor* allele: forward, CTCAGTAGTGGTATGTGCTCAG; reverse, GGGTA-CAGTATGTCAGCACAG. The control mice for *Raptor* LKO were *Rptor*^fl/fl mice that were administered with GFP adenovirus at a similar titer as the Cre-expressing adenovirus. For generation of liver-specific *Mtor* knockout mice, *Mtor*^fl/fl mice were crossed with *Alb*-Cre mice. PCR genotyping of *Mtor* LKO mice was performed with the following primers to detect the 3′ LoxP site of the targeted *Mtor* allele: forward, TTATGTTTGATA ATTGCAGTTTTGGCTAGCAGT; reverse, TTTAGGACTCCTTCTGTGACATACATTTCCT.

**Mouse experiments**. Mice were housed in colony cages with a 12 h light/12 h dark cycle (lights on at 7:00 a.m. as ZT0, lights off at 7:00 p.m. as ZT12) in a temperature-controlled environment with free access to food and water. All of the experiments were conducted on 8–12-week-old male mice. To overexpress specific proteins in the liver, adenoviruses ($5 \times 10^8$ pfu per mouse) were injected through the tail vein. If needed, mice were fasted overnight from ZT12 to ZT4, then refed for 2 h and killed at ZT6. Animals were maintained with all relevant ethical regulations for animal testing and research. All animal experiments were approved by the Animal Care and Use Committee at Tsinghua University.

**Metabolic studies and histology**. Blood glucose values were determined using a LifeScan automatic glucometer. Triglyceride (TR0100, Sigma) and cholesterol (10007640, Cayman) levels in the liver and plasma, plasma insulin (10-1247-01, Mercodia), and plasma-free fatty acid levels (294-636, Wako) were measured according to the manufacturer's instructions. To extract bile acids from liver tissues, 100 mg tissue was homogenized in 75% ethanol. The homogenate was incubated at 50 °C for 2 h and centrifuged at $6000 \times g$ for 10 min. Total bile acid levels in plasma and liver tissues were determined using a Total Bile Acids Kit (Diazyme, DZ092A) and normalized with the tissue weight of each sample. For tissue histology, mouse tissues were fixed in 4% paraformaldehyde (PFA) and paraffin embedded. Sections (5 µm) were used for hematoxylin and eosin staining.

**Indirect calorimetry, physical activity, and food intake**. Energy expenditure and food intake were simultaneously measured for individually housed mice with a PhenoMaster system (TSE Systems). Mice were allowed to acclimatize in the chambers for at least 24 h. Food and water were provided ad libitum in the appropriate devices and measured by the built-in automated instruments. Relative fat mass was measured with an EchoMRI analyzer.

**Hyperinsulinemic-euglycemic clamps**. The assays were performed as previously reported[49,50]. After mice were anesthetized, catheters (PE-10) were implanted in the right jugular vein and tunneled subcutaneously and exteriorized behind the neck. Mice were allowed to recover for 4–5 days before the clamp procedure. Mice were fasted overnight, then weighed to calculate the insulin dose and placed in a restrainer. An equilibration syringe was connected to the catheter and an initial bolus of [6,6-$^2$H]glucose (600 µg kg$^{-1}$) was administered, followed by continuous infusion of [6,6-$^2$H]glucose (30 µg kg$^{-1}$ min$^{-1}$). Isotopic enrichment was achieved at ~90 min after the onset of isotope infusion, and blood samples for isotope measurements were collected at $t = -10$ and 0 min. Following this basal infusion period, insulin was constantly infused at the rate of 6 mU kg$^{-1}$ min$^{-1}$ until termination of the study. [6,6-$^2$H]glucose was infused together with glucose at various rates until the blood glucose concentration reached a constant level of about $100 \pm 5$ mg dl$^{-1}$. Blood samples were taken at $t = 110$ and 120 min.

All plasma samples (15 µl) were deproteinized by gently mixing them with 60 µl cold methanol (pre-chilled at −80 °C) and incubating for 1–2 h at −80 °C. The samples were then centrifuged with $14,000 \times g$ at 4 °C for 10 min. The supernatant was transferred to a new tube and lyophilized to produce a pellet.

High-resolution MS (Q Exactive, Thermo) coupled with ultimate 3000 ultra-high performance liquid chromatography (UPLC) was used for the analysis. GDR was calculated as constant isotope infusion rate divided by enrichment of isotopes. HGP and GDR were calculated in the basal state and during the steady-state portion of the clamp. Tracer-determined rates were quantified using Steele's equation. Under steady-state conditions, the rate of glucose disappearance, or total GDR, is equal to the sum of the rate of endogenous glucose production (HGP) and the exogenous GIR. The IS-GDR is equal to the total GDR minus the basal GDR.

**Plasmids and adenoviruses**. HA-tagged TJP1 was amplified from pEGFP-C1 ZO1 (Addgene, 30313) and cloned into pcDNA3 vector containing a HA tag. Site-directed mutagenesis was performed using a PCR-based strategy with PfuUltra II Fusion HS DNA polymerase (Agilent). FLAG-PER1 and FLAG-CRY1 were kindly provided by Dr. Erquan Eric Zhang (National Institute of Biological Sciences, Beijing). HA-S6K was previously described[31]. FLAG-PER1 (1-1147) was amplified from FLAG-PER1 and cloned into p3 × FLAG-CMV-7.1 vector. V5-tagged BMAL1 and CLOCK were from CCSB-Broad Lentiviral Expression Library (GE Dharmacon). FLAG-tagged mTOR was amplified from myc-mTOR (Addgene, 1861) and cloned into p3 × FLAG-CMV-7.1 vector. All adenoviruses were generated using the pAdTrack/pAdEasy or pShuttle/pAdEasy system and amplified in HEK293 cells. Adenoviruses were purified by CsCl gradient centrifugation and dialyzed in 1 × PBS with 15% glycerol. GFP adenovirus has been described previously[31]. All expressed constructs used in this study were confirmed by sequencing.

**Cell culture**. HEK293T (ATCC) and MDCK II cells (ATCC) were cultured in DMEM supplemented with 10% FBS (HyClone), 100 mg ml$^{-1}$ penicillin–streptomycin and maintained at 37 °C and 5% CO$_2$. Mouse primary hepatocytes were isolated and cultured as previously described[31]. For cell synchronization, cells were exposed to dexamethasone (0.1 µM) with or without 250 nM Torin1 and then harvested at different time points. For amino acid starvation and stimulation of cells, cells were washed twice with 1 × PBS, incubated overnight in amino acid-free RPMI1640 and stimulated for 30 min with the amino

acid mixture (Sigma, M5550) added directly to the amino acid-free RPMI1640. Cells were processed for biochemical or immunofluorescence assays as described below. When Torin1 was used, cells were incubated in 250 nM Torin1 for 1 h before amino acid stimulation. All cell lines were routinely tested for mycoplasma using a PCR detection kit (Sigma, MP0035).

Collagen gel sandwich cultures were prepared by spreading 2 ml of collagen solution (BD Biosciences, at 1 mg ml$^{-1}$ in DMEM) over 10 cm dishes and allowing the gel layer to dry in an incubator for 2 h. Hepatocytes ($3 \times 10^6$) were plated in collagen-coated 10 cm dishes. Hepatocytes were allowed to attach for 2 h in M199 supplemented with 2% FBS, 0.2% BSA and 100 mg ml$^{-1}$ penicillin–streptomycin. Cultures were aspirated and then the top layer of collagen gel was added. After drying for 2 h, the medium was replaced with M199 supplemented with 100 mg ml$^{-1}$ penicillin–streptomycin.

**Subcellular fractionation**. Hepatocytes or freshly harvested livers (100 mg) were placed in a glass homogenizer containing 1 ml cold lysis buffer (10 mM Tris–HCl, pH 8.0, 10 mM NaCl, 0.2% NP-40) and then dounced 30 times. The homogenates were incubated on ice for 15 min, and 200 µl was saved as the total cell lysate. The nuclear fraction was pelleted by centrifugation at $10,000 \times g$ for 10 min at 4 °C and resuspended in 800 µl RIPA buffer (25 mM Tris–HCl pH 7.4, 1% NP-40, 1% sodium deoxycholate, 0.1% SDS, 150 mM NaCl), then sonicated and centrifuged at $12,000 \times g$ for 10 min at 4 °C. The supernatant was saved as the cytosolic fraction.

To evaluate the distribution of TJP1 in lysosome fractions, the lysosomes in cells were extracted using a Lysosome Isolation Kit (Sigma, LYSISO1) according to the manufacturer's instructions. After density gradient centrifugation of the cytosolic fraction, five bands were visible in the tube. These fractions were withdrawn using long tips starting from the top of the gradient, and number (with the top fraction as no. 1). Fractions numbered 1–4 were treated with 8 mM calcium chloride solution on ice for 15 min to precipitate the rough endoplasmic reticulum and mitochondria, and then centrifuged at $5000 \times g$ for 15 min at 4 °C. After this step, most of the lysosomes remained in the supernatant.

To test the distribution of TJP1 at junctions or in the cytoplasm, cells were incubated in ice-cold cell lysis buffer with 1% Triton X-100 for 15 min on ice. Cells were then harvested gently using a scraper and centrifuged at $10,000 \times g$ for 10 min at 4 °C. The supernatant was labeled as the soluble fraction. The cell pellets were resuspended in ice-cold RIPA buffer and pipetted gently to disperse the pellets. This suspension was centrifuged at $10,000 \times g$ for 10 min at 4 °C, and the supernatant was labeled as the insoluble fraction.

**Transepithelial electrical resistance**. The TER assay was performed as previously described[51]. In brief, the primary hepatocytes in collagen gel sandwich cultures and MDCK II cells were plated into Transwell™ filters (Corning, CLS3450-24EA) at confluence and allowed to attach for 2 days. TER was measured with a Millipore electrical resistance system (Millipore, MERS00002). The resulting TER values were calculated after subtracting the blank value (no cells), and were documented as Ω cm$^2$.

**IP, immunoblot, and immunostaining**. Assays were performed as previously described[31,52]. For co-IP experiments, TJP1-overexpressing cells were collected in cell lysis buffer (150 mM NaCl, 50 mM HEPES pH 7.4, 1% Triton X-100). HA-tagged TJP1 was immunoprecipitated with HA-beads (Thermo 26182). mCherry-tagged TJP1 and mutants were immunoprecipitated by anti-mCherry antibody (Origene, TA180028). For endogenous co-IP experiments, proteins from cells or tissues were extracted in cell lysis buffer. The endogenous TJP1 was immunoprecipitated by anti-TJP1 antibody (Invitrogen 61-7300) coupled with Protein A/G Plus Agarose (Thermo 26161). All immunoprecipitated samples were washed at least four times with lysis buffer. For all endogenous IP experiments, rabbit IgG was used as a negative control.

For immunoblotting, cells or mouse tissues were homogenized in cell lysis buffer. Protein concentrations were determined using the BCA Protein Assay Kit (Thermo Fisher, 23227). Samples were loaded on SDS–PAGE gels and then transferred to nitrocellulose membranes. Immunoblotting was done in gelatin buffer (50 mM Tris–HCl pH 7.4, 150 mM NaCl, 5 mM EDTA, 0.05% Tween-20) with the corresponding antibodies. The antibodies were purchased as follows: anti-PER1 (A8449, 1:1000), anti-PER2 (A13168, 1:1000), anti-CRY1 (A6890, 1:1000), anti-CLDN1 (A2196, 1:1000), anti-LAMP1 (A2582, 1:1000), and anti-LMNA (A0249, 1:1000), ABclonal; anti-actin (66009, 1:5000), anti-BMAL1 (14268-1-AP, 1:1000), anti-Rev-Erbα (14506-1-AP, 1:1000), anti-Cingulin (21369-1-AP, 1:1000), and anti-V5 (66007-1-Ig, 1:1000), Proteintech; Rabbit anti-TJP1 (61-7300, 1:1000), mouse anti-TJP1 (339100, 1:1000) and anti-TJP2 (374700, 1:1000), Thermo Fisher; Rat anti-TJP1 (MABT11, 1:1000), Millipore; anti-CLOCK (5157S, 1:1000), anti-S6K (2708, 1:1000), anti-pS6K (9234, 1:1000), anti-mTOR (2983, 1:1000), anti-RAPTOR (2280T, 1:1000), anti-EEA1 (3288, 1:1000), anti-TSC1 (6935S, 1:1000), and anti-TJP3 (3704, 1:1000), Cell Signaling Technology; anti-KDEL (ab12223, 1:1000), anti-COX IV (ab33985, 1:1000) and anti-TGN46 (ab50595, 1:1000), Abcam; anti-mCherry (TA180028, 1:1000), Origene; anti-LAMP1 (L1418, 1:1000), anti-Tubulin (T6199, 1:10000), anti-FLAG (F1804, 1:5000), Sigma-Aldrich; anti-GFP (MMS-118P, 1:5000), anti-HA (MMS-101P, 1:1000), COVANCE; anti-His

(D291-3, 1:1000), MBL. The phospho-S1614 TJP1 antibody (1:1000) was generated and purified by Beijing Prorevo Biotech Co. Ltd.

For immunofluorescence assays, cells or liver tissues were fixed in 4% formaldehyde solution. Fixed cells or 8 μM-thick frozen sections of liver were blocked in PBS with 0.2% Triton X-100 and 5% BSA. Samples were stained overnight with specific primary antibodies in blocking solution at 4 °C, washed three times in TBST and then incubated at room temperature with fluorescent dyes in blocking solution for 1 h. Samples were incubated in DAPI solution for 10 min to stain the DNA, and then coverslipped. Tiled images were obtained from an epifluorescence microscope (Zeiss) and the exposure time for each channel was kept constant for all slides. Signal intensity was quantified using ImageJ.

**Transmission electron micrographs (TEM)**. Liver tissues were fixed with 2.5% glutaraldehyde in PBS by perfusion, then fixed in 2.5% glutaraldehyde and 2% PFA for 16 h. Tissues were post-fixed in 1% osmium tetroxide for 1 h and then dehydrated through an ethanol series. After changing the transitional solvent several times, tissues were embedded in Spurr's resin. Ultra-thin sections were stained with uranyl acetate and lead citrate, then analyzed using a transmission electron microscope (Tecnai Spirit 120 kV).

**Protein expression and purification**. Fusion proteins were purified as previously reported[50]. Briefly, His-HA-tagged TJP1, TJP1 mutant (924-1748aa) and His-PER1 fusion proteins were purified from Mtor knockout primary hepatocytes after 48-h adenoviral infection. His-HA-tagged TJP1 mutants (1-510aa, 511-923aa) and His-CRY1 were purified from E. coli. Cells or bacteria were collected, resuspended in buffer containing 25 mM Tris pH 8.0, 150 mM NaCl and protease inhibitor, sonicated and centrifuged at $16,000 \times g$ for 10 min. The supernatant was then centrifuged at $300,000 \times g$ for 60 min. After centrifugation at $300,000 \times g$ for 30 min, the supernatant was applied to nickel affinity resin (Ni-NTA, QIAGEN). The eluent was purified by size-exclusion chromatography (Superdex 200, GE Healthcare) in buffer (25 mM HEPES pH 7.4, 50 mM KCl, 5 mM MgCl₂, and 5 mM MnCl₂). The peak fractions were pooled and concentrated for subsequent analysis.

**In vitro kinase assay**. HA-tagged TJP1, TJP1/6A, and S6K fusion proteins were purified from Mtor knockout primary hepatocytes after 48-h infection. The kinase assay was performed as reported[31]. The reaction system (20 μl), containing 150 ng fusion protein, 20 ng truncated mTOR (Millipore, 14-770) in reaction buffer (25 mM HEPES pH 7.4, 50 mM KCl, 5 mM MgCl₂, and 5 mM MnCl₂), 50 μM cold ATP and 2 μCi [γ-³²P]ATP was incubated for 30 min at 30 °C. Reactions were stopped by adding 5 μl sample buffer, then boiled for 10 min and analyzed by SDS–PAGE followed by detection with phospho-specific antibodies.

**RNA extraction and qPCR**. Total RNA from cells or mouse tissues was extracted using a Total RNA Purification kit (GeneMark). cDNA was obtained using the RevertAid First Strand cDNA Synthesis kit (Thermo). RNA levels were measured with the LightCycler 480 II (Roche) as previously described[31,52]. The following primers were used for qPCR:

Actin-forward: GTCCACCCCGGGGAAGGTGA
Actin-reverse: AGGCCTCAGACCTGGGCCATT
Bmal1-forward: GTAGATCAGAGGGCGACAGC
Bmal1-reverse: CCTGTGACATTCTGCGAGGT
Clock-forward: AGCGATGTCTCAAGCTGCAA
Clock-reverse: TGCATGGCTCCTAACTGAGC
Cry1-forward: CACTGGTTCCGAAAGGGACTC
Cry1-reverse: CTGAAGCAAAAATCGCCACCT
Dbp-forward: TCTAGGGACACACCCAGTCC
Dbp-reverse: ATGGCCTGGAATGCTTGACA
Rev-erbα-forward: TGGCATGGTGCTACTGTGTAAGG
Rev-erbα-reverse: ATATTCTGTTGGATGCTCCGGCG
Per1-forward: TGAAGCAAGACCGGGAGAG
Per1-reverse: CACACACGCCATCACATCAA
Per2-forward: GAAAGCTGTCACCACCATAGAA
Per2-reverse: AACTCGCACTTCCTTTTCAGG

**Mass spectrometry**. To identify TJP1-interacting proteins, primary hepatocytes were infected with HA-TJP1 adenovirus. Immunoprecipitates of HA-TJP1 were prepared for MS studies as previously reported[31,50] and analyzed by electrospray ionization tandem MS on a Thermo LTQ Orbitrap instrument. To identify the mTOR phospho-site (s) in PER1 and CRY1, primary hepatocytes were infected with FLAG-PER1 or FLAG-CRY1 adenovirus and treated with or without 250 nM Torin1 for 2 h. Immunoprecipitates of FLAG-PER1 and FLAG-CRY1 were used to determine phospho-site (s) by MS. The MS proteomics data have been deposited to the ProteomeXchange Consortium via the PRIDE[53] partner repository with the dataset identifier PXD016917 and PXD016919.

**Statistical methods**. Age-matched and weight-matched male mice were randomly assigned for the experiments. The animal numbers used for all experiments are stated in the corresponding figure legends. No animals were excluded from statistical analyses, and the investigators were not blinded in the studies. All studies were performed on at least three independent occasions. Results are reported as mean ± s.e.m. Comparison of different groups was carried out using two-tailed unpaired Student's t-test, one-way ANOVA or two-way ANOVA. Differences were considered statistically significant at $P < 0.05$. No statistical methods were used to predetermine sample size.

**Reporting summary**. Further information on research design is available in the Nature Research Reporting Summary linked to this article.

## Data availability
All relevant data supporting the key findings of the study are available within the article and its Supplementary Information files or from the corresponding author upon reasonable request. The proteomics data are available via ProteomeXchange with identifier PXD016917 and PXD016919. The source data underlying Figs. 1–7 and Supplementary Figs. 1–6 are provided as a Source Data file.

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

## Acknowledgements

We thank Drs. Erquan Eric Zhang, Ying Xu, Yi Liu, and Suneng Fu for critical discussion and technical help. This work was supported by grants from the Ministry of Science and Technology of the People's Republic of China (2017YFA0503404 and 2016YFC1304803) and the National Natural Science Foundation of China (31625014, 31621063, 31830040, and 91957206).

## Author contributions

Y.L. Y.Z., and Y.W. designed the study and analyzed the data; Y.L., Y.Z., T. L. and J.H. performed experiments; Y.L. performed all the experiments during revision; Y.W. wrote the paper. All authors reviewed and commented on the manuscript.

## Competing interests

The authors declare no competing interests.
