## [Peer Review File · Nature Communications]

Reviewers' comments:

Reviewer #1 (Remarks to the Author):

The circadian timing system of mammals is a complex, interacting network of individual, cell-based oscillators. While the phase of the master clock located in the Suprachiasmatic nuclei (SCN) is affected by the external light-dark phase, most of the peripheral oscillators are affected by other systemic cues such as feeding or temperature rhythms. The authors investigated here an important, still unanswered question, how is the liver circadian oscillator affected by feeding? They provide a series of well-controlled experiments but overall the entire story is not yet properly meshed together. I see two areas where the story has to be improved:

1) More exhaustive explanation of the experiments.

The manuscript is very tight and much information (e.g., regarding the displayed items) is missing or misleading.

1a) Abstract: '...is entrained by fasting-feeding cycles to maintain self-sustained oscillation.' It is known in the field that the self-sustained oscillations are not affected by the fasting-feeding cycles but the phase of oscillation is affected, e.g. by comparing nighttime with daytime feeding.

1b) same for page 3, introduction, it should read: '...the phase of the circadian clock in the liver is entrained...'

1c) page 5, rescue experiment: you use two constructs in Fig. 1E that become explained only in Fig. 2.

1d) page 6: 'Strikingly, amino acid...' Why suddenly use amino acids, please explain rationale.

1e) Fig. S3C: why use LAMP1 as a marker and what does this assay indicate?

1f) In general, the discussion should be improved to make it clearer to the general reader.

1g) please indicate in the figure legends, how many data points were compared and which statistical tests were applied. Also, for the many western blots quantification of the bands should be performed.

1h) Please check, whether actin is really a stable normalization gene to be used in the qPCR. It is obvious that the cell shape is different in SD- and SD+ cultures, which may partially be based on the actin cytoskeleton.

2) Provide an experiment for feeding-entrainment of the circadian oscillator affected by the supposed mechanism.

From point 1a, I am not convinced that the authors mean the same thing as I, when they refer to (fasting-) feeding entrainment of the hepatic circadian clock. Historically, the impact of feeding on the hepatic circadian oscillator was shown by comparing daytime with nighttime feeding (citations 5 and 7). Under such experimental conditions, the phase of the hepatic circadian oscillator became shifted by up to 12h but the phase of the SCN clock remained unaffected. A dampening of the circadian oscillation by feeding has, to my best knowledge, not been observed. This is referred to as the feeding entrainment of the hepatic circadian oscillator. To address this issue, the knock-out mice used in this study have simply to be fed for two weeks either during the day or the night, and then the phases of the liver circadian oscillators have to be compared.

Jürgen A. Ripperger, PhD

Reviewer #2 (Remarks to the Author):

This is an interesting paper describing a new function of ZO-1/TJP1 in the regulation of circadian gene expression and suggesting that it functions as an mTOR effector in this process.

The paper is a largely carefully performed study. However, there are some unconvincing and/or confusing observations reported that should be addressed as the molecular and cellular mechanism by which ZO-1 functions in this process remain ill-defined.

Major

1) Fig. 1: Why is only the amplitude affected but not the rhythm as such? If control of the repressor is switched off, should one not expect an effect on the rhythm as well?

2) Page 5: I would not be able to judge whether tight junction formation is affected based on the images in Figure S1. That would require a much more careful analysis. As the authors state, there are systems in which ZO-1 deletion does not (greatly) affect tight junction formation. However, it does in

other systems (e.g., PLoS One. 2014 Aug 26;9(8):e104994.). There are certainly a number of papers claiming functional effects of ZO-1 knockdown/knockout on tight junctions.

I am unable to identify any tight junctions in the provided EM images in Fig. S1E; the quality of the images is inadequate.

Measuring permeability properties with a more sensitive assay would be required.

3) Figure 2B: The co-localization data are odd. They only place with co-localization is an unidentified structure in the cytoplasm and not at junctions where ZO-1 is normally localized. I am not aware of ZO-1 having been reported to localize to such a structure and this paper shows this only by overexpression of fluorescently tagged protein, suggesting it may be an artefact. Where do ZO-1 and PER1 co-localize in wild type cells? Where does this interaction occur? Do phosphorylation site mutations affect recruitment of ZO-1 to these structures or to tight junctions?

4) The mTOR interaction co-immunoprecipitation is only based on overexpressed proteins. Can this interaction be detected endogenously and where does it occur (see point 3)?

5) The spread of mutations in ZO-1 affecting PER1 binding is surprising as they are distributed over 70% of a very large protein. How are these individual mutations affecting PER1 nuclear translocation? Do the mutations affect ZO-1 localization to the cytoplasmic structure where the interaction with PER1 appears to occur?

Minor

6) Page 5: 'Interestingly, SD+ culture dramatically enhanced the expression of Rev-Erba and circadian amplitude (Figure 1A), suggesting that the junctions or junction proteins may have a critical effect on circadian rhythm.'

Considering the difference in the culture system, I don't see how a junction-specific statement could be made.

7) Page 5: 'However, Tjp1 deficiency notably repressed the expression of E-box-containing genes (Rev-Erba, Dbp and Per1) in the liver, but not in the SCN,'

Why should it affect the SCN if the knockout is liver-specific? This statement is misleading.

Reviewer #3 (Remarks to the Author):

In this article, the authors aim to identify the molecular mechanisms by which feeding entrains the hepatic clock. They show that the tight junction protein 1 (TJP1) interacts with the circadian factor Period 1 (PER1) in a process that prevents PER1's nuclear translocation. The authors propose that, during feeding time, mTOR phosphorylates TJP1 which, in turn, attenuates TJP1's association to PER1. The authors propose this process favors nuclear shuttling and dampens circadian oscillation.

The paper is well-reasoned and presented. In general, the biochemical aspect of the article is fine (see details below) but there is substantial redundant information and some inconsistencies that need to be addressed. It also misses the opportunity of delivering a major contribution by not expanding the work beyond the interaction of three molecules.

Major points

a- In Fig. 1, the authors conclude "these results demonstrate that TJP1 modulates hepatic circadian amplitude by inhibiting PER1/CRY1 nuclear translocation." However, this conclusion overstates the results shown in Fig. 1 and should be completely reformulated to better reflect the results presented. For example, the authors do not show CRY1 shows the same profile of translocation as PER1 in TJP1 LKO cells and at different circadian times. This needs to be shown.

b- In Fig. 1, the analysis of PER2 mRNA/protein/translocation is conspicuously absent. This is particularly surprising when considering all other core circadian genes, except the one that might have a function redundant to that of PER1, were analyzed.

c- Page 6: The conclusion of Fig. 2A/B should be tempered. The sentence "This indicates that mTOR promotes the dissociation of TJP1 and PER1" should be presented in the context of the involvement of mTOR activity to influence the stability of the complex. Fig. 2B should also show the localization of CRY1 and the experiment should also be performed in the presence of Torin 1 alone and in combination with AA.

d- Fig. 2C: what are the levels of CRY1 detected in the IP experiment at ZT6 and ZT18? What happened with the levels of PER1/CRY1 in ZT6 and ZT18 in TJP1 LKO animals? What is the activity of mTOR in LKO animals at those times? At what times the animals receive any feeding in this experiment?

e- Fig. 2D: Please add a control experiment in which transfected cells are treated with Torin before proceeding to immunoprecipitate FLAG-mTOR and detect TJP1 binding.

f- Fig. 3C: A blot showing the level of expression of Rev-Erba in the nuclear/cytosol fraction should be added. This is largely to support that mTOR-TJP1-PER1 disruption and translocation results in the corresponding increase in Rev-Erba protein level.

g- Page 8: the sentence “In summary, these results showed that both TJP1 and mTOR are critical for the feeding-entrained circadian clock in the liver.” This conclusion is again an overstatement as “entrainment” is not shown. The authors look at localization of PER1 under each fasting/feeding condition and in ONLY ONE time point.

h- Fig. S4C: this figure is troublesome in the context of what it is presented in Fig. S4A-B. Why is the amount of PER1 in the nucleus of wild type fasted and fed animals the same when the IF result shows a clear difference? Why is the amount of PER1 in the nucleus of fasted wild type and Tjp1 LKO and Tsc1 KO animals comparable when the IF shows a different result? NOTE: surprisingly the cytosol results are not shown but are needed to truly evaluate the distribution of PER1 from one compartment to another.

i- Figs. 4A-C do not add any new information to what it is shown in Fig. S4 in terms of the model and again Cry1 needs to be shown and the blot with the cytosolic fractions displayed to make a convincing point about protein redistribution.

j- The authors miss a tremendous opportunity of showing the relevance of the TJP1-PER1 signaling pathway during the fasting/feeding cycle by not including some metabolic results using the various animal strains they have on hand and, specially, considering such a clear cellular phenotype.

k- Fig. S4E: the model presented includes CRY and no data on CRY binding is shown in this article. In the model, the authors propose mTOR phosphorylates TJP1 despite being bound to PER1, this result has not been shown in the paper. It is not clear from the data presented in the article if mTOR phosphorylation precedes or occurs after PER1 binding/dissociation to TJP1. The model needs to be revisited to better reflect the data in the article.

Minor points

a- The abbreviation mTOR is not defined in the “abstract” section.

b- Under the “Tjp1 knockout promotes nuclear translocation of PER1 and(page 5)” there is no description of what the “TJP1/6A or 6P” constructs are or their purpose on the main text (fig. 1E-1G).

Reviewer #1 (Remarks to the Author):

The circadian timing system of mammals is a complex, interacting network of individual, cell-based oscillators. While the phase of the master clock located in the Suprachiasmatic nuclei (SCN) is affected by the external light-dark phase, most of the peripheral oscillators are affected by other systemic cues such as feeding or temperature rhythms. The authors investigated here an important, still unanswered question, how is the liver circadian oscillator affected by feeding? They provide a series of well-controlled experiments but overall the entire story is not yet properly meshed together. I see two areas where the story has to be improved:

1) More exhaustive explanation of the experiments. The manuscript is very tight and much information (e.g., regarding the displayed items) is missing or misleading.

1a) Abstract: '...is entrained by fasting-feeding cycles to maintain self-sustained oscillation.' It is known in the field that the self-sustained oscillations are not affected by the fasting-feeding cycles but the phase of oscillation is affected, e.g. by comparing nighttime with daytime feeding.

We appreciate the reviewer's comments and we have amended the text in the revised manuscript.

1b) same for page 3, introduction, it should read: '...the phase of the circadian clock in the liver is entrained...'

We appreciate the reviewer's comments and we have amended the text at Page 3 of the revised manuscript.

1c) page 5, rescue experiment: you use two constructs in Fig. 1E that become explained only in Fig. 2.

In the revised manuscript, we have reorganized the data (Figure 3), and these constructs are not used in Figure 1.

1d) page 6: 'Strikingly, amino acid...' Why suddenly use amino acids, please explain rationale.

We appreciate the reviewer's comments and we have included the description (Page 6).

1e) Fig. S3C: why use LAMP1 as a marker and what does this assay indicate?

Recruitment of mTOR to lysosomes is required for mTOR activation¹. In SD+ culture, mTOR is more dispersed and has less colocalization with LAMP1 (a marker of lysosomes) than that in SD- culture. These results partially explain why mTOR is less active in SD+ culture than in SD- culture (Figures S4B and S4C in the revised manuscript).

1f) In general, the discussion should be improved to make it clearer to the general reader.

We appreciate the reviewer's comments and we have improved the discussion in the revised manuscript.

1g) please indicate in the figure legends, how many data points were compared and which statistical tests were applied. Also, for the many western blots quantification of the bands should be performed.

We appreciate the reviewer's comments and we have included the statistical information in the figure legends and the quantitation results of immunoblots in the revised manuscript.

1h) Please check, whether actin is really a stable normalization gene to be used in the qPCR. It is obvious that the cell shape is different in SD- and SD+ cultures, which may partially be based on the actin cytoskeleton.

The Cp value of actin qPCR is 14.58 ± 0.31 for WT/SD-, 14.75 ± 0.24 for *Tjp1* KO/SD-, 15.47 ± 0.29 for WT/SD+ and 15.36 ± 0.36 for *Tjp1* KO/SD+. Although there is a slight difference for actin mRNA between SD- and SD+ cultures, there is no obvious difference between WT and *Tjp1* KO cells.

2) Provide an experiment for feeding-entrainment of the circadian oscillator affected by the supposed mechanism.

From point 1a, I am not convinced that the authors mean the same thing as I, when they refer to (fasting-) feeding entrainment of the hepatic circadian clock. Historically, the impact of feeding on the hepatic circadian oscillator was shown by comparing daytime with nighttime feeding (citations 5 and 7). Under such experimental conditions, the phase of the hepatic circadian oscillator became shifted by up to 12h but the phase of the SCN clock remained unaffected. A dampening of the circadian oscillation by feeding has, to my best knowledge, not been observed. This is referred to as the feeding entrainment of the hepatic circadian oscillator. To address this issue, the knock-out mice used in this study have simply to be fed for two weeks either during the day or the night, and then the phases of the liver circadian oscillators have to be compared.

We appreciate the reviewer's comments and we have amended the text in the revised manuscript. In this manuscript, we have studied the acute effect of feeding (at one time point) on the hepatic circadian clock. We fasted mice from ZT12 to ZT4 and then re-fed them for another two hours to test the role of the mTOR-TJP1-PER1 axis in the hepatic circadian clock.

Reviewer #2 (Remarks to the Author):

This is an interesting paper describing a new function of ZO-1/TJP1 in the regulation of circadian gene expression and suggesting that it functions as an mTOR effector in this process.

The paper is a largely carefully performed study. However, there are some unconvincing and/or confusing observations reported that should be addressed as the molecular and cellular mechanism by which ZO-1 functions in this process remain ill-defined.

Major

1) *Fig. 1: Why is only the amplitude affected but not the rhythm as such? If control of the repressor is switched off, should one not expect an effect on the rhythm as well?*

Our results showed that *Tjp1* LKO promotes the nuclear translocation of PER1/CRY1 at ZT6 but not at ZT18, indicating that TJP1 affects the circadian clock at specific circadian times. Our subsequent results in this paper further showed that TJP1 modulates circadian amplitude as a downstream mediator of mTOR during refeeding after fasting. These results indicate that the mTOR-TJP1 axis provides an additional layer to modulate the circadian clock at the posttranslational level, but the regulatory loop of the circadian clock at the transcriptional level is not blocked. Therefore, TJP1 mainly affects circadian amplitude in the liver.

2) *Page 5: I would not be able to judge whether tight junction formation is affected based on the images in Figure S1. That would require a much more careful analysis. As the authors state, there are systems in which ZO-1 deletion does not (greatly) affect tight junction formation. However, it does in other systems (e.g., PLoS One. 2014 Aug 26;9(8):e104994.). There are certainly a number of papers claiming functional effects of ZO-1 knockdown/knockout on tight junctions. I am unable to identify any tight junctions in the provided EM images in Fig. S1E; the quality of the images is inadequate. Measuring permeability properties with a more sensitive assay would be required.*

In the revised manuscript, we have provided new evidence to evaluate tight junction formation by measurement of transepithelial electrical resistance (TER)^{2, 3}. As shown in Figure S1E, TER is comparable in WT and *Tjp1* LKO hepatocytes. TEM pictures with high resolution are also included in the revised manuscript. Compared to the acute effect of *Tjp1* knockdown on tight junction formation, it is possible that TJP2 compensates for the deficiency of TJP1 in *Tjp1* LKO mice since they have redundant function. In addition, WT and *Tjp1* LKO mice had similar CLDN1 staining, hepatic bile acid contents and plasma bile acid levels, which further confirms that *Tjp1* LKO has similar tight junctions (Figures S1A and S1F-S1H). Together, these results indicate that *Tjp1* LKO does not affect tight junction formation.

3) *Figure 2B: The co-localization data are odd. They only place with co-localization is an unidentified structure in the cytoplasm and not at junctions where ZO-1 is normally localized. I am not aware of ZO-1 having been reported to localize to such a structure and this paper shows this only by overexpression of fluorescently tagged protein, suggesting it may be an artefact. Where do ZO-1 and PER1 co-localize in wild type cells? Where does this interaction occur?*

Do phosphorylation site mutations affect recruitment of ZO-1 to these structures or to tight junctions?

Both endogenous TJP1 and overexpressed TJP1 have a clear localization at junctions, and also in the perinuclear region, where TJP1 colocalizes with LAMP1 (a marker of lysosomes) (Figures 2B and S3A). Although the function of lysosome-localized TJP1 is unclear, it is reported that lysosome-dependent redistribution and degradation of TJP1 contributes to hyperglycemia-increased blood-brain barrier permeability⁴.

Under amino acid starvation, TJP1 colocalizes with PER1 at lysosomes, while activated mTOR disrupts this association and promotes the nuclear translocation of PER1 (Figures 2B, 2C and S3).

No, phosphorylation site mutations do not affect recruitment of TJP1 to the lysosome and/or tight junctions (Figure S3A).

4) The mTOR interaction co-immunoprecipitation is only based on overexpressed proteins. Can this interaction be detected endogenously and where does it occur (see point 3)?

Yes, this interaction can be detected for both endogenous and overexpressed TJP1 and mTOR at the lysosome (Figures 2E, S2I and S2J).

5) The spread of mutations in ZO-1 affecting PER1 binding is surprising as they are distributed over 70% of a very large protein. How are these individual mutations affecting PER1 nuclear translocation? Do the mutations affect ZO-1 localization to the cytoplasmic structure where the interaction with PER1 appears to occur?

No, phosphorylation site mutations do not affect recruitment of TJP1 to the lysosome and/or tight junctions (Figure S3A). However, nuclear translocation of PER1 stimulated by amino acids in primary hepatocytes was blocked by the phosphorylation-defective mutant of TJP1 (6A), while the phosphorylation-mimic mutant of TJP1 (6D) constitutively promotes nuclear translocation of PER1 (Figure S3).

Minor

6) Page 5: 'Interestingly, SD+ culture dramatically enhanced the expression of Rev-Erba and circadian amplitude (Figure 1A), suggesting that the junctions or junction proteins may have a critical effect on circadian rhythm.'

Considering the difference in the culture system, I don't see how a junction specific statement could be made.

We appreciate the reviewer's comments and have removed this sentence.

7) Page 5: 'However, Tjp1 deficiency notably repressed the expression of E-box-containing genes (Rev-Erba, Dbp and Per1) in the liver, but not in the SCN,'

Why should it affect the SCN if the knockout is liver-specific? This statement is misleading.

We have modified the statement and added further explanation at Page 5.

Reviewer #3 (Remarks to the Author):

In this article, the authors aim to identify the molecular mechanisms by which feeding entrains the hepatic clock. They show that the tight junction protein 1 (TJP1) interacts with the circadian factor Period 1 (PER1) in a process that prevents PER1's nuclear translocation. The authors propose that, during feeding time, mTOR phosphorylates TJP1 which, in turn, attenuates TJP1's association to PER1. The authors propose this process favors nuclear shuttling and dampens circadian oscillation.

The paper is well-reasoned and presented. In general, the biochemical aspect of the article is fine (see details below) but there is substantial redundant information and some inconsistencies that need to be addressed. It also misses the opportunity of delivering a major contribution by not expanding the work beyond the interaction of three molecules.

Major points

a- In Fig. 1, the authors conclude “these results demonstrate that TJP1 modulates hepatic circadian amplitude by inhibiting PER1/CRY1 nuclear translocation.” However, this conclusion overstates the results shown in Fig. 1 and should be completely reformulated to better reflect the results presented. For example, the authors do not show CRY1 shows the same profile of translocation as PER1 in TJP1 LKO cells and at different circadian times. This needs to be shown.

We appreciate the reviewer's comments and have shown the CRY1 data in the revised manuscript (Figures 1C, 1E, 1F and 1G). Similar to PER1 nuclear translocation, Tjp1 LKO promotes CRY1 nuclear translocation.

b- In Fig. 1, the analysis of PER2 mRNA/protein/translocation is conspicuously absent. This is particularly surprising when considering all other core circadian genes, except the one that might have a function redundant to that of PER1, were analyzed.

We appreciate the reviewer's comments and have included the analysis of PER2 mRNA/protein/translocation (Figures 1C, 1F, 1G and S1I). Tjp1 LKO has no dramatic effect on PER2 nuclear translocation.

c- Page 6: The conclusion of Fig. 2A/B should be tempered. The sentence “This indicates that mTOR promotes the dissociation of TJP1 and PER1” should be presented in the context of the involvement of mTOR activity to influence the stability of the complex. Fig. 2B should also show the localization of CRY1 and the experiment should also be performed in the presence of Torin 1 alone and in combination with AA.

We have amended the description at Page 7 of the revised manuscript.

The cellular localization of PER1 and CRY1 were performed in primary hepatocytes treated with or without amino acids in the presence or absence of Torin1 (Figures 2B, 2C, S2D, S2E and S3). Amino acid treatment promotes PER1/CRY1 nuclear translocation, while Torin1 pretreatment blocks the AA-stimulated effect.

d- Fig. 2C: what are the levels of CRY1 detected in the IP experiment at ZT6 and ZT18? What happened with the levels of PER1/CRY1 in ZT6 and ZT18 in TJP1 LKO animals? What is the activity of mTOR in LKO animals at those times? At what times the animals receive any feeding in this experiment?

We performed the co-IP assays in liver extracts from *ad lib*-fed WT and *Tjp1* LKO mice at ZT6 and ZT18. Both PER1 and CRY1 can be pulled down by anti-TJP1 antibody in liver extracts from WT mice but not *Tjp1* LKO mice (Figure 2D). *Tjp1* LKO has no effect on PER1/CRY1 protein levels and mTOR activation (Figures 1C and 2D).

e- Fig. 2D: Please add a control experiment in which transfected cells are treated with Torin before proceeding to immunoprecipitate FLAG-mTOR and detect TJP1 binding.

We have performed this experiment using both endogenous and overexpressed TJP1 and mTOR in the presence or absence of Torin1 (Figures 2E and S2J). Torin1 treatment decreased the TJP1:mTOR association.

f- Fig. 3C: A blot showing the level of expression of Rev-Erba in the nuclear/cytosol fraction should be added. This is largely to support that mTOR-TJP1-PER1 disruption and translocation results in the corresponding increase in Rev-Erba protein level.

A blot of Rev-Erb α , a nuclear localized nuclear receptor, is included in the revised manuscript. The changes in the Rev-Erb α protein levels are similar to the changes in its mRNA level (Figures 4B-4D in the revised manuscript).

g- Page 8: the sentence "In summary, these results showed that both TJP1 and mTOR are critical for the feeding-entrained circadian clock in the liver." This conclusion is again an overstatement as "entrainment" is not shown. The authors look at localization of PER1 under each fasting/feeding condition and in ONLY ONE time point.

We have amended the description at Page 9 of the revised manuscript.

*h- Fig. S4C: this figure is troublesome in the context of what it is presented in Fig. S4A-B. Why is the amount of PER1 in the nucleus of wild type fasted and fed animals the same when the IF result shows a clear difference? Why is the amount of PER1 in the nucleus of fasted wild type and *Tjp1* LKO and *Tsc1* KO animals comparable when the IF shows a different result? NOTE: surprisingly*

the cytosol results are not shown but are needed to truly evaluate the distribution of PER1 from one compartment to another.

The IF results showed the mCherry-PER1, while the blots showed the endogenous PER1. Although PER1 antibody is fine for detecting endogenous PER1 on blots, it cannot detect endogenous PER1 by immunostaining. In addition, cytoplasm-localized mCherry-PER1 is smeared. Therefore, the signal intensity of cytoplasm-localized mCherry-PER1 looks weaker than the nuclear-localized mCherry-PER1.

We have included the PER1 blots in cytosolic fractions in the revised manuscript (Figure 5C). *Tjp1* LKO or *Tsc1* LKO enhanced nuclear translocation of PER1 with a concomitant decrease of cytosolic PER1, while *Raptor* LKO mice showed a higher amount of cytosolic PER1 and a lower amount of nuclear PER1.

i- Figs. 4A-C do not add any new information to what it is shown in Fig. S4 in terms of the model and again Cry1 needs to be shown and the blot with the cytosolic fractions displayed to make a convincing point about protein redistribution.

Fig. S4 (Fig. 5 in the revised manuscript) showed that both mTOR and TJP1 are critical for refeeding-induced PER1 nuclear translocation, while Figs 4A-C (Fig. 6 in the revised manuscript) showed that TJP1, as a downstream mediator of mTOR, is critical for refeeding-induced PER1 nuclear translocation.

We have included the blots of CRY1 in cytosolic fractions in the revised manuscript (Figures 5C, 6B and 6G). The results showed that CRY1 has a similar redistribution to PER1.

j- The authors miss a tremendous opportunity of showing the relevance of the TJP1-PER1 signaling pathway during the fasting/feeding cycle by not including some metabolic results using the various animal strains they have on hand and, specially, considering such a clear cellular phenotype.

We appreciate the reviewer's comments and we have included the metabolic results. *Tjp1* LKO mice fed with a regular diet (RD) showed an enhanced nuclear translocation of PER1 and insulin sensitivity, and reduced *Rev-Erb α* expression, fasting blood glucose level, plasma insulin level, plasma triglyceride (TG) level and hepatic TG accumulation. In addition, *Tjp1* LKO restored the high fat diet (HFD)-induced metabolic profile and enhanced insulin sensitivity.

k- Fig. S4E: the model presented includes CRY and no data on CRY binding is shown in this article. In the model, the authors propose mTOR phosphorylates TJP1 despite being bound to PER1, this result has not been shown in the paper. It is not clear from the data resented in the article if mTOR phosphorylation precedes or occurs after PER1 binding/dissociation to TJP1. The model needs

to be revisited to better reflect the data in the article.

We appreciate the reviewer's comments and we have revised the model (Figure 8).

Minor points

a- The abbreviation *mTOR* is not defined in the "abstract" section.

We have defined this abbreviation in the revised manuscript.

b- Under the "Tjp1 knockout promotes nuclear translocation of *PER1* and(page 5)" there is no description of what the "TJP1/6A or 6P" constructs are or their purpose on the main text (fig. 1E-1G).

We have included the description in the revised manuscript.

REFERENCES

1. Saxton RA, Sabatini DM. mTOR Signaling in Growth, Metabolism, and Disease. *Cell* **169**, 361-371 (2017).
2. Srinivasan B, Kolli AR, Esch MB, Abaci HE, Shuler ML, Hickman JJ. TEER measurement techniques for in vitro barrier model systems. *J Lab Autom* **20**, 107-126 (2015).
3. Wang Y, *et al.* Tyrosine phosphorylated Par3 regulates epithelial tight junction assembly promoted by EGFR signaling. *EMBO J* **25**, 5058-5070 (2006).
4. Zhang S, An Q, Wang T, Gao S, Zhou G. Autophagy- and MMP-2/9-mediated Reduction and Redistribution of ZO-1 Contribute to Hyperglycemia-increased Blood-Brain Barrier Permeability During Early Reperfusion in Stroke. *Neuroscience* **377**, 126-137 (2018).

Reviewers' comments:

Reviewer #1 (Remarks to the Author):

The authors identified here an interaction between the Tight junction protein 1 (TJP1) and the circadian repressor protein Period 1 (PER1). This interaction is affected by mTOR signaling to integrate the feeding status. On the other hand, the interaction affects the nuclear translocation of PER1 and CRY1 into the nucleus to modulate the amplitude of circadian gene expression in the liver. The story is interesting and mostly conclusive. The activation of mTOR by refeeding disrupts the interaction of TJP1 and PER1, which together with CRY1 enters into the nucleus to repress circadian gene expression. However, there are some confusing facts in the present form of the manuscript.

A) Figure 6. Fasting/refeeding experiment: why would mTOR phosphorylate TJP1 and why would TJP1 interact with PER1 at this time point? According to Figure S2F mTOR activity is the lowest at ZT6, i.e. the time when the experiment was performed and in general there is very few PER1 at this time point. We have to conclude that mTOR becomes activated in response to feeding. This conjecture has to be verified by comparing the accumulation/ activity of PER1, TJP1 and mTOR in ad libitum fed, fasted and refed animals. I prefer to see this on one panel with the same conditions and exposure times/quantification.

B) Figures 7. Why would there be a difference between the nuclear import of PER1 in normal chow- and high fed diet-fed Tjp1 LKO animals? The effect of the deficiency should be exactly the same, if the simple model is correct?

C) Another important issue is the lack of phosphorylation of PER1 and CRY1 by mTOR. Unfortunately, in Figure S2G there is nothing to see concerning the phosphorylation of either protein, because it is a simple western blot. On the other hand, the mass spectrometry analysis is not exhaustive and one cannot exclude that all peptides were recovered. The Ramanathan et al., paper (Ref 26) suggested an impact of mTOR on CRY1 accumulation, hence a direct effect in this context should be ruled out. A classical in vitro kinase experiment with ³²P and purified proteins should do the job. By the way, a characterization of the phospho-specific TJP1 antibody is missing and it appears that only one of them was used throughout the manuscript?

Reviewer #3 (Remarks to the Author):

The authors have addressed all my previous experimental concerns satisfactory and redefined the proposed model to be more in line with the findings reported.

Just a minor comment, please replace the word “novel” in the last sentence of the abstract by “previously uncharacterized.”

I have no objection to the publication of this manuscript.

Reviewer #4 (Remarks to the Author):

This is an interesting study, that reveals a new implication of ZO-1 in a signaling mechanism involving regulation of a transcription factor which controls circadian rhythm in the liver. However, the data about ZO-1 leave something to desire with respect to the clarification of the topological basis (localization) of the functional interaction with mTOR, the biochemical basis of the interaction, and the implication of additional ZO-1-dependent mechanisms.

Regarding the major comments by Reviewer 2.

Comment n. 1 was addressed satisfactorily.

Comment n. 2 was partially addressed. The TEM figure is too small, one cannot see anything. The TER values are so low that they are barely meaningful. Labeling for additional TJ proteins (occludin, cingulin, etc) would be useful.

Comment n. 3. I agree with reviewer n. 2 that the lysosomal localization of ZO-1 is quite odd and not satisfactorily documented. There is diffuse labeling all over the cytoplasm, and lysosomal localization must be demonstrated by clearer data, also in additional cell types, with different ZO-1 antibodies, using different fixations, and in cells depleted or KO for ZO-1. Biochemical evidence (co-IP with lysosomal markers, fractionation) should also be provided. What about the possibility, instead, that mTOR localizes to junctions? The labeling by mTOR antibodies (Figure S2-I) is not very clear. How were these antibodies validated? One could speculate that lysosome-associated ZO-1 is not so strongly associated with the cytoskeleton, so perhaps immunoprecipitation could be done on cytoskeleton-associated and soluble pools, to show that only “soluble” (non junction-associated) ZO-1 associated with mTOR? Does mTOR promote the junctional accumulation of ZO-1?

Comment n. 4 was addressed satisfactorily.

Comment n. 5 was addressed partially. The quality of the immunofluorescence labeling and the fact that exogenous (not-normalized in levels) proteins were expressed does not allow to draw conclusions on the effect of the ZO-1 mutations on the localization. However, these would be interesting with respect to the mechanisms by which ZO-1 regulates PER1, for example if this is by a phosphorylation-dependent shift from the lysosome-associated to the junction associated pool. These experiments should be done in the background of ZO-1-KO cells, and not only in hepatocytes, but in other experimental cellular models, with a careful normalization of junctional vs lysosomal localizations using internal reference standards for each structure.

Additional comments.

1. The biochemical basis for the ZO-1-PER1 interaction should be clarified. After all, the whole regulatory mechanism is centered on the interaction between PER-1 and ZO-1, and on the modulation of the interaction. However, this is not characterised in sufficient detail. The authors use only co-IP assays, which do not demonstrate direct interaction. In vitro experiments using purified proteins and protein fragments must be carried out, especially to map the region of the proteins involved in the interaction, and to determine precisely how phosphomutants/phosphomimetics affect the interaction: is this because the phosphorylation sites are in the binding region? Or is it because they affect ZO-1 conformation? Data from the Turner and Citi laboratories, about the role of different ZO-1 protein domains in the junctional/cytoplasmic localizations, and in the conformation of ZO-1 should not be ignored.

2. ZO-1 has been implicated in the junctional recruitment of regulators of Rho GTPases, and oscillations in actin dynamics have been linked to circadian rhythms (Gerber et al Cell 2013). Have the authors tested whether drugs that interfere with Rho GTPases and their effectors, and/or the actin cytoskeleton modulate PER1 translocation to the nucleus?

Reviewer #1 (Remarks to the Author):

The authors identified here an interaction between the Tight junction protein 1 (TJP1) and the circadian repressor protein Period 1 (PER1). This interaction is affected by mTOR signaling to integrate the feeding status. On the other hand, the interaction affects the nuclear translocation of PER1 and CRY1 into the nucleus to modulate the amplitude of circadian gene expression in the liver. The story is interesting and mostly conclusive. The activation of mTOR by refeeding disrupts the interaction of TJP1 and PER1, which together with CRY1 enters into the nucleus to repress circadian gene expression. However, there are some confusing facts in the present form of the manuscript.

A) Figure 6. Fasting/refeeding experiment: why would mTOR phosphorylate TJP1 and why would TJP1 interact with PER1 at this time point? According to Figure S2F mTOR activity is the lowest at ZT6, i.e. the time when the experiment was performed and in general there is very few PER1 at this time point. We have to conclude that mTOR becomes activated in response to feeding. This conjecture has to be verified by comparing the accumulation/ activity of PER1, TJP1 and mTOR in ad libitum fed, fasted and refed animals. I prefer to see this on one panel with the same conditions and exposure times/quantification.

We appreciate the reviewer's comments and we have performed the experiments as the reviewer suggested. As shown in Reviewer Figure 1 (below), a 2 hr-refeeding after 16 hr fasting dramatically enhances mTOR activity (evaluated by pS6K protein levels) and slightly increases PER1 accumulation, but has no effect on TJP1 accumulation. Although mTOR activity is lower in *ad lib*-fed mice at this time point, refeeding dramatically enhances mTOR activity and thereby phosphorylates TJP1 and further affects the TJP1:PER1 interaction.

B) Figures 7. Why would there be a difference between the nuclear import of PER1 in normal chow- and high fed diet-fed Tjp1 LKO animals? The effect of the deficiency should be exactly the same, if the simple model is correct?

A high-fat diet not only increases mTOR activity but also modulates other factors, such as casein kinase or phosphatases, to affect PER1 cellular localization^{1,2}. In addition, it is possible that TJP1 is one of the downstream effectors of mTOR to regulate PER1 cellular localization. The difference between the nuclear import of PER1 in normal chow- and high-fat diet-fed *Tjp1* LKO mice suggests that multiple signaling pathways may be involved in the regulation of PER1 nuclear translocation.

C) Another important issue is the lack of phosphorylation of PER1 and CRY1 by mTOR. Unfortunately, in Figure S2G there is nothing to see concerning the phosphorylation of either protein, because it is a simple western blot. On the other hand, the mass spectrometry analysis is not exhaustive and one cannot exclude that all peptides were recovered. The Ramanathan et al., paper (Ref 26) suggested an impact of mTOR on CRY1 accumulation, hence a direct effect in this context should be ruled out. A classical in vitro kinase experiment with 32P and purified proteins should do the job.

We appreciate the reviewer's comments and we have performed the *in vitro* kinase assay. No obvious phosphorylation of PER1 and CRY1 by mTOR was observed (Figure S3H), indicating that PER1 and CRY1 are not the direct substrates of mTOR.

By the way, a characterization of the phospho-specific TJP1 antibody is missing and it appears that only one of them was used throughout the manuscript?

A characterization of the phospho-specific TJP1 antibody was performed by immunoblotting (Figure S3M), which confirmed the specificity of this antibody. Although we designed six phospho-specific TJP1 antibodies targeting different phospho-sites, there is only one pTJP1 (S1617) antibody which can recognize the phosphorylated endogenous TJP1.

Reviewer #3 (Remarks to the Author):

The authors have addressed all my previous experimental concerns satisfactory and redefined the proposed model to be more in line with the findings reported.

Just a minor comment, please replace the word "novel" in the last sentence of the abstract by "previously uncharacterized."

I have no objection to the publication of this manuscript.

We have revised the text according to the reviewer's advice and we thank the reviewer for her/his support for publication.

Reviewer #4 (Remarks to the Author):

This is an interesting study, that reveals a new implication of ZO-1 in a signaling mechanism involving regulation of a transcription factor which controls circadian rhythm in the liver. However, the data about ZO-1 leave something to desire with respect to the clarification of the topological basis (localization) of the functional interaction with mTOR, the biochemical basis of the interaction, and the implication of additional ZO-1-dependent mechanisms.

Regarding the major comments by Reviewer 2.

Comment n. 1 was addressed satisfactorily.

We thank the reviewer for her/his support.

Comment n. 2 was partially addressed. The TEM figure is too small, one cannot see anything. The TER values are so low that they are barely meaningful. Labeling for additional TJ proteins (occludin, cingulin, etc) would be useful.

We have labeled Cingulin (CGN) to test the effect of *Tjp1* KO on tight junction formation in mouse primary hepatocytes or liver tissues (Figure S1A and S1F). Consistent with the results of CLDN1 staining and TER measurement, *Tjp1* KO hepatocytes have similar tight junctions to wildtype hepatocytes.

Comment n. 3. I agree with reviewer n. 2 that the lysosomal localization of ZO-1 is quite odd and not satisfactorily documented. There is diffuse labeling all over the

cytoplasm, and lysosomal localization must be demonstrated by clearer data, also in additional cell types, with different ZO-1 antibodies, using different fixations, and in cells depleted or KO for ZO-1. Biochemical evidence (co-IP with lysosomal markers, fractionation) should also be provided.

We appreciate the reviewer's comments and we have confirmed the lysosomal localization of TJP1 in primary hepatocytes and MDCK cells by immunostaining and immunoblots of lysosome fraction (Figure 2B, Figure S3A, S3C and Reviewer Figure 2A-2E, below). We used one rabbit anti-TJP1 polyclonal antibody (antigen 463-1109 aa of human TJP1, Cat# 61-7300, Thermo Fisher) and one mouse monoclonal antibody (antigen 334-634 aa of human TJP1, Cat# 339100, Thermo Fisher) to perform the assays, both of which can specifically detect TJP1 (Reviewer Figure 2A, below).

TJP1 has a lysosomal localization in both primary hepatocytes and MDCK cells evaluated by TJP1 staining after either PFA or methanol fixation, while *Tjp1* KO or KD abolished these signals (Figure 2B, Figure S3A and Reviewer Figure 2B-2D, below). In addition, by immunoblotting, TJP1 shows a similar distribution pattern with LAMP1, a lysosomal marker, but not EEA1 (early endosome marker), KDEL (ER marker), TGN46 (Golgi marker) and COXIV (mitochondrial marker) (Figure S3C and Reviewer Figure 2E, below).

Together, these results indicate that TJP1 has a lysosomal localization.

What about the possibility, instead, that mTOR localizes to junctions? The labeling by mTOR antibodies (Figure S2-I) is not very clear. How were these antibodies validated?

mTOR antibody (Cat#2983, Cell Signaling Technology) has been validated for immunostaining and immunoblotting by more than 800 published papers from different labs. Our results also showed that this antibody specifically recognizes mTOR in both immunoblotting and immunostaining (Reviewer Figure 3, below). mTOR is diffuse in the cytoplasm in the absence of amino acids, while it accumulates at lysosomes after amino acid treatment.

mTOR localizes to non-junction fractions (soluble fractions) in the presence or absence of amino acid treatment in both primary hepatocytes and MDCK cells (Figure S3D and Reviewer Figure 2F, below). This is further confirmed by mTOR staining in mouse primary hepatocytes (Figure S3I).

One could speculate that lysosome-associated ZO-1 is not so strongly associated with the cytoskeleton, so perhaps immunoprecipitation could be done on cytoskeleton-associated and soluble pools, to show that only "soluble" (non junction-associated) ZO-1 associated with mTOR? Does mTOR promote the junctional accumulation of ZO-1?

TJP1 exists in both Triton X-100 soluble and insoluble fractions, while mTOR is only in soluble fractions (Figure S3D and Reviewer Figure 2F, below). The mTOR:TJP1 association was confirmed in soluble fractions (Figure 2E).

Amino acid treatment enhances mTOR accumulation in the lysosomal fraction and activates mTOR (Figure S3C and Reviewer Figure 2E, below). Activated mTOR does not affect TJP1 localization in the lysosomal fraction or at tight junctions (Figure S3D and Reviewer Figure 2F, below).

Comment n. 4 was addressed satisfactorily.

We are grateful for the reviewer's support.

Comment n. 5 was addressed partially. The quality of the immunofluorescence labeling and the fact that exogenous (not-normalized in levels) proteins were expressed does not allow to draw conclusions on the effect of the ZO-1 mutations on the localization. However, these would be interesting with respect to the mechanisms by which ZO-1 regulates PER1, for example if this is by a phosphorylation-dependent shift from the lysosome-associated to the junction associated pool. These experiments should be done in the background of ZO-1-KO cells, and not only in hepatocytes, but in other experimental cellular models, with a careful normalization of junctional vs lysosomal localizations using internal reference standards for each structure.

Both wildtype TJP1 and TJP1 mutants have similar distribution patterns in lysosomal fractions and junctions, as evaluated by immunostaining and immunoblots (Figure S4 and Reviewer Figure 2G-2H, below).

Additional comments.

1. The biochemical basis for the ZO-1-PER1 interaction should be clarified. After all, the whole regulatory mechanism is centered on the interaction between PER-1 and ZO-1, and on the modulation of the interaction. However, this is not characterised in sufficient detail. The authors use only co-IP assays, which do not demonstrate direct interaction. In vitro experiments using purified proteins and protein fragments must be carried out, especially to map the region of the proteins involved in the interaction, and to determine precisely how phosphomutants/phosphomimetics affect the interaction: is this because the phosphorylation sites are in the binding region? Or is it because they affect ZO-1 conformation? Data from the Turner and Citi laboratories, about the role of different ZO-1 protein domains in the junctional/cytoplasmic localizations, and in the conformation of ZO-1 should not be ignored.

We appreciate the reviewer's comments and we have performed the binding assays. Co-IP assays in HEK293T cells show that the C-terminus (511-1748 aa) of TJP1 is critical for the TJP1:PER1 interaction (Figure S2D). *In vitro* binding experiments using purified proteins further confirmed these results (Figure S2E). These results show that TJP1 directly interact with PER1.

In fact, the identified mTOR phosphorylation sites in TJP1 are localized within the region of TJP1 that interacts with PER1, and some of the phosphorylation sites are also located in the protein domains of TJP1. Based on previous reports^{3, 4}, it is possible that the phosphorylation of TJP1 directly affects TJP1:PER1 binding, or changes the conformation of TJP1 to modulate TJP1:PER1 binding. More detailed structural information is required to answer this question.

2. ZO-1 has been implicated in the junctional recruitment of regulators of Rho GTPases, and oscillations in actin dynamics have been linked to circadian rhythms (Gerber et al Cell 2013). Have the authors tested whether drugs that interfere with Rho GTPases and their effectors, and/or the actin cytoskeleton modulate PER1 translocation to the nucleus?

We appreciate the reviewer's comments and we have tested the effect of cytoskeletal modulators on PER1 nuclear translocation. As shown in Reviewer Figure 4 (below), Cytochalasin D (CytoD), Latrunculin B (LatB), Rho activator or Rho inhibitor treatment has no effect on PER1 nuclear translocation, although they affect the actin cytoskeleton.

REFERENCES

1. Virshup DM, Eide EJ, Forger DB, Gallego M, Harnish EV. Reversible protein phosphorylation regulates circadian rhythms. *Cold Spring Harb Symp Quant Biol* **72**, 413-420 (2007).
2. Cunningham PS, Ahern SA, Smith LC, da Silva Santos CS, Wager TT, Bechtold DA. Targeting of the circadian clock via CK1delta/epsilon to improve glucose homeostasis in obesity. *Sci Rep* **6**, 29983 (2016).
3. Spadaro D, et al. Tension-Dependent Stretching Activates ZO-1 to Control the Junctional Localization of Its Interactors. *Curr Biol* **27**, 3783-3795 e3788 (2017).
4. Odenwald MA, et al. ZO-1 interactions with F-actin and occludin direct epithelial polarization and single lumen specification in 3D culture. *J Cell Sci* **130**, 243-259 (2017).

Reviewer Figure 1: Effect of feeding or fasting on mTOR activity.

A-D, Immunoblots (**A**) and statistical results showing the effect of *ad lib* feeding, fasting (16 hr) or refeeding 2 hr after 16-hr fasting on the levels of PER1 (**B**), TJP1 (**C**) and pS6K (**D**) in liver extracts from mice.

NS, no statistical significance. Data are shown as mean \pm s.e.m. Comparison of different groups was carried out using one-way ANOVA. * $P < 0.05$, ** $P < 0.01$, *** $P < 0.001$, $n = 6$.

Reviewer Figure 2: Characterization of TJP1 cellular localization.

A, Immunoblots showing the expression of TJP1 in WT or *Tjpr1*^{-/-} (KO) primary hepatocytes. (M) indicates mouse anti-TJP1 (339100, Thermo Fisher), while (R) indicates rabbit anti-TJP1 (61-7300, Thermo Fisher).

B, Cellular localization of TJP1 in mouse primary hepatocytes after paraformaldehyde (PFA) or methanol fixation. Scale bars, 10 μ m.

C, Evaluation of *Tjpr1* knockdown stable cell lines. The expression of TJP1 in four cell lines stably expressing different targeted *Tjpr1* sequences were tested by immunoblotting.

D, Cellular localization of TJP1 in MDCK II cells after paraformaldehyde (PFA) or methanol fixation. Scale bars, 10 μ m.

E, Immunoblots showing the distribution of TJP1 in lysosome fractions. The red asterisk indicates LAMP1. MDCK II cells incubated with amino acid-free RPMI1640 for 6 hrs were exposed to amino acids for 30 min. TCL, total cell lysate.

F, Immunoblots showing the relative distribution of TJP1 in the junction fraction (Triton X-100 insoluble) and cytosolic fraction (Triton X-100 soluble). MDCK II cells incubated with amino acid-free RPMI1640 for 6 hrs were exposed to amino acids for 30 min.

G, Immunoblots showing the distribution of WT TJP1 and TJP1 mutants in lysosome fractions of MDCK II cells. The red asterisk indicates LAMP1. TCL, total cell lysate.

H, Immunoblots showing the relative distribution of WT TJP1 and TJP1 mutants in the junction fraction (Triton X-100 insoluble) and cytosolic fraction (Triton X-100 soluble) of MDCK II cells.

Reviewer Figure 3: Evaluation of mTOR antibody.

A-B, Immunoblots (**A**) and immunostaining (**B**) showing the specificity of mTOR antibody. KD indicates knockdown of mTOR in HEK293T cells. Scale bar, 10 μ m.

Reviewer Figure 4: Effect of cytoskeleton modulators on cellular localization of PER1.

A-B, Cellular localization of GFP-PER1 (**A**) and statistical analysis of the results (**B**) showing the effect of cytoskeleton modulators on cellular localization of PER1 in mouse primary hepatocytes cultured in a collagen sandwich configuration. Mouse primary hepatocytes incubated with amino acid-free RPMI1640 for 6 hrs were treated with 2 μ M Cytochalasin D (CytoD) or 1 μ M Latrunculin B (LatB) for 2 hrs, or treated with 1 ng μ l⁻¹ Rho activator II or Rho inhibitor for 4 hrs before phalloidin staining. Scale bar, 10 μ m.

REVIEWERS' COMMENTS:

Reviewer #1 (Remarks to the Author):

The authors have addressed my recommendations.

Reviewer #4 (Remarks to the Author):

The Authors have addressed my comments.

Reviewer #1 (Remarks to the Author):

The authors have addressed my recommendations.

We thank the reviewer's support.

Reviewer #4 (Remarks to the Author):

The Authors have addressed my comments.

We thank the reviewer's support.